# A therapeutic approach to pantothenate kinase associated neurodegeneration

Lalit Kumar Sharma[1,2,4], Chitra Subramanian[1], Mi-Kyung Yun[3], Matthew W. Frank[1], Stephen W. White [3], Charles O. Rock [1], Richard E. Lee [2] & Suzanne Jackowski [1]

Pantothenate kinase (PANK) is a metabolic enzyme that regulates cellular coenzyme A (CoA) levels. There are three human *PANK* genes, and inactivating mutations in *PANK2* lead to pantothenate kinase associated neurodegeneration (PKAN). Here we performed a library screen followed by chemical optimization to produce PZ-2891, an allosteric PANK activator that crosses the blood brain barrier. PZ-2891 occupies the pantothenate pocket and engages the dimer interface to form a PANK•ATP•Mg$^{2+}$•PZ-2891 complex. The binding of PZ-2891 to one protomer locks the opposite protomer in a catalytically active conformation that is refractory to acetyl-CoA inhibition. Oral administration of PZ-2891 increases CoA levels in mouse liver and brain. A knockout mouse model of brain CoA deficiency exhibited weight loss, severe locomotor impairment and early death. Knockout mice on PZ-2891 therapy gain weight, and have improved locomotor activity and life span establishing pantazines as novel therapeutics for the treatment of PKAN.

[1] Department of Infectious Diseases, St. Jude Children's Research Hospital, Memphis, TN 38105, USA. [2] Department of Chemical Biology and Therapeutics, St. Jude Children's Research Hospital, Memphis, TN 38105, USA. [3] Department of Structural Biology, St. Jude Children's Research Hospital, Memphis, TN 38105, USA. [4]Present address: Nurix, Inc, 1700 Owens Street, Suite 205, San Francisco, CA 94158, USA. Correspondence and requests for materials should be addressed to S.J. (email: suzanne.jackowski@stjude.org)

A rare, life-threatening neurological disorder known as pantothenate kinase-associated neurodegeneration (PKAN) arises from mutations in the human *PANK2* gene[1] leading to a prominent extrapyramidal movement disorder and a characteristic deposition of iron in the basal ganglia[2–4]. Pantothenate kinase (PANK, EC 2.7.1.33) is the first and rate-controlling step in the only pathway for coenzyme A (CoA) biosynthesis. CoA is a major acyl group carrier in biology and participates as a key cofactor and regulator of intermediary metabolism[5]. Three genes express four closely-related mammalian isoforms: PANK1α, PANK1β, PANK2, and PANK3[5]. The isoform expression level and potent feedback inhibition by acyl-CoAs control the intracellular CoA content[6–10]. The physiological evaluation of PANK knockout mice and mice treated with a PANK inhibitor establish a key role for the intracellular CoA concentration in supporting oxidative metabolism, ketone utilization and glucose homeostasis[11–14]. PKAN pathology is thought to arise from neuronal CoA deficiency. This view is strengthened by the recent discovery that mutations in CoA synthase cause a similar neurodegenerative disease[15]. The *PANK2* gene is abundant in human neuronal tissues and the majority of the mutations associated with PKAN result in the expression of truncated or mutant PANK2 proteins with little or no catalytic activity[16]. Human PANK2 is localized to the mitochondrial intermembrane space[17,18]. Mouse PanK2 was reported to be mitochondrial[19,20], but a mouse PanK2 mitochondrial targeting sequence has not been identified, and others report a cytosolic localization[18,21]. *Pank2*−/− mice do not recapitulate the neurological PKAN phenotype[13,22].

There are no disease-modifying treatments for PKAN[4], and therapeutics are desperately needed. One approach is to use protected phosphopantothenates to bypass PANK and elevate CoA[23]. Phosphopantetheine, CoA, or S-acetyl-phosphopantetheine have been suggested as PKAN modulators based on their ability to reverse hopantenate inhibition of CoA synthesis[24–30]. All these approaches use compounds with physicochemical properties that suggest they will not penetrate the blood brain barrier[31,32], and protected phosphopantothenates have no effect on mouse brain CoA levels[23].

Here, we report the development of a drug capable of allosterically activating the alternate PANK isoforms as a potential PKAN therapeutic. The lead pantazine, PZ-2891, arose from the LipE-guided chemical optimization of a hit from a high-throughput screen designed to identify inhibitors and activators of PANK3[33,34]. Due to the high cooperativity of the PANK dimer[9] the binding of PZ-2891 to one protomer locks the opposite protomer in a constitutively active state that is refractory to feedback inhibition by acyl-CoA. PZ-2891 crosses the blood brain barrier to elevate brain CoA. A mouse model of CoA deficiency employing the neuron-selective deletion of PANK1 and PANK2 genes exhibits weight loss, severely impaired locomotor activity, and early death. PZ-2891-treated knockout mice gain weight, and have improved locomotor activity and life span. These data suggest pantazines as a novel approach to treating PKAN.

## Results

**LipE-guided optimization identifies PZ-2891 as a PANK modulator.** Prior attempts at hit-to-lead optimization of hits from a large high throughput screen for modulators of PANK3 failed to generate suitable chemical leads due to flat structure-activity relationships and poor solubility[34]. To address these shortcomings, we reevaluated the hit list using the alternative approach of filtering for compounds with both lead-like molecular weight (<350) and lipophilic ligand efficiency (LipE > 2). LipE ($pIC_{50}$ − cLogP) blends both potency and lipophilicity to quantitatively compare the molecules, and ranks them based on physicochemical properties suitable for biophysical characterization[35,36]. These efforts prioritized a piperazine urea hit, PZ-2789 (Fig. 1a), which was subjected to a hit-to-lead optimization using LipE as the primary driving metric to create a compound series called pantazines (Fig. 1a). PANK activation in cells (see below) requires a high-affinity pantazine, and a PANK inhibition assay without acetyl-CoA was used to rank the pantazines. The structure–activity relationships revealed the importance of a small branched alkyl group at $R_1$, a carbonyl (H-bond acceptor) next to the piperidine ring, and an electron

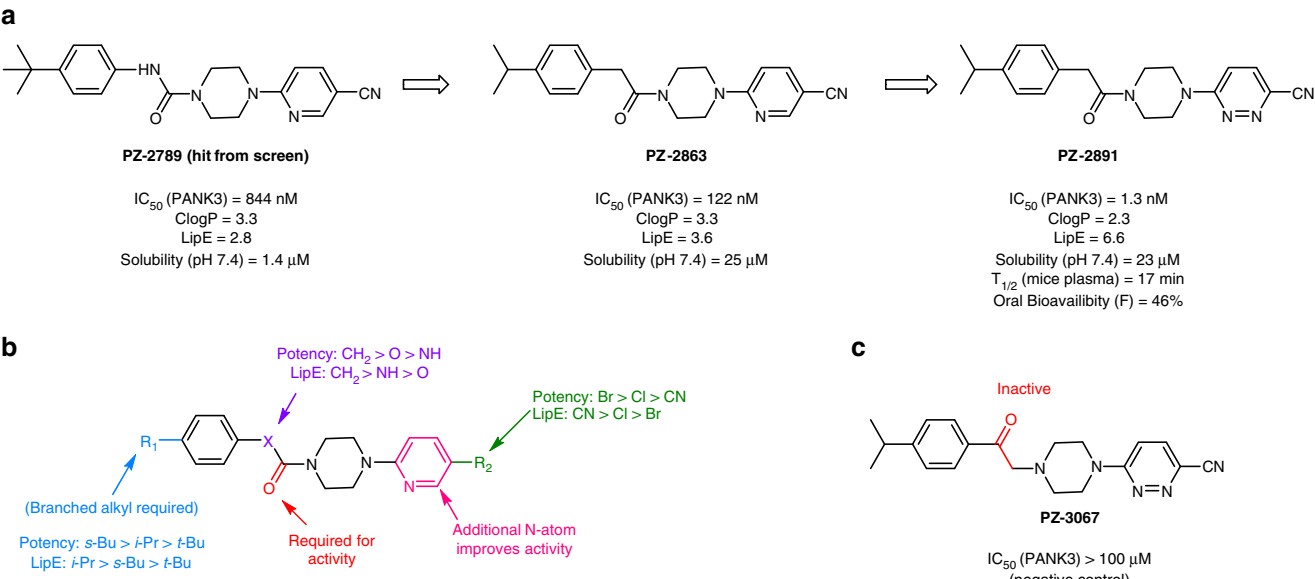

**Fig. 1** Chemical progression of pantazines. **a** Lipophilic ligand efficiency (LipE = $pIC_{50}$ – clogP) guided optimization of the pantazine series from the initial hit (PZ-2789) to PZ-2891. **b** Summary of the structure-activity relationships of >60 compounds synthesized during the optimization of the pantazine scaffold. **c** Chemical structure of an inactive pantazine, PZ-3067, which is used as a negative control in cellular assays

withdrawing group at $R_2$ for the activity of the compounds (Fig. 1b). The LipE-guided optimization culminated in the identification of PZ-2891 that was ∼800 times more potent than the initial hit. The improved potency and reduced lipophilicity (∼1 log unit reduction in cLogP) resulted in a significantly improved LipE for PZ-2891 (Fig. 1a). The three key changes from the initial hit to PZ-2891 were: replacement of the t-butyl group with isopropyl, switching the urea linker with an acetamide linker, and substitution of the nicotinonitrile side chain with pyridazine-3-carbonitrile. Compared to the screening hit, PZ-2891 exhibited superior adsorption, distribution, metabolism, and excretion properties that indicated it could be used for proof of principle experiments in cells and animals (Fig. 1a; Supplementary Tables 1, 2). PZ-3067 (Fig. 1c), an inactive isomer of PZ-2891, was selected as a biochemical control compound for mechanism of action studies (Supplementary Table 1).

**Pantazines bind to the PANK3•ATP•Mg$^{2+}$ complex.** The lead pantazine, PZ-2891, inhibited PANK3 with nM affinity, whereas the inactive PZ-3067 had no effect (Fig. 2a). PZ-2891 inhibited all three human and mouse pantothenate kinase isoforms (Supplementary Fig. 1). We used a less potent pantazine, PZ-2724 (IC$_{50}$ = 1.1 μM) to determine how pantazines interrupted the ordered kinetic mechanism of PANK3 (Supplementary Fig. 2). Kinetic analysis of PANK3 with respect to ATP showed that PZ-2724 was an uncompetitive inhibitor resulting in a decreased Vmax and ATP Km consistent with the pantazines binding to the PANK3•ATP•Mg$^{2+}$ complex. Pantazine inhibition with respect to pantothenate was noncompetitive. The conclusion that pantazines bind tightly to the PANK3•ATP•Mg$^{2+}$ complex was corroborated by thermal stabilization studies showing that PZ-2891 stabilized PANK3 only in the presence of ATP (Fig. 2b). Both ATP and ATP + PZ-2891 stabilization of PANK3 to thermal denaturation were concentration-dependent and saturable, whereas PZ-2891 alone was not (Supplementary Fig. 3). Also, [$^3$H]ATP completely dissociated from PANK3 during gel filtration chromatography, but when PZ-2891 was added to the sample, the PANK3•[$^3$H]ATP•Mg$^{2+}$•PZ-2891 complex was stabilized and isolated (Fig. 2c). Analysis of the PANK3•ATP•Mg$^{2+}$•PZ-2891 complex stability by surface plasmon resonance showed a long residence time (34 min), and a calculated affinity of 0.2 nM (Fig. 2d).

**PZ-2891 binds across the PANK dimer interface to stabilize the active PANK conformation.** The crystal structure of the PANK3•AMPPNP•Mg$^{2+}$•PZ-2891 complex (Supplementary Table 3; Supplementary Fig. 4) reveals that PZ-2891 binds across the dimer interface to simultaneously interact with both PANK3 protomers (Fig. 2e). The 'ring of ligands' linking the two protomers explains the high thermal stability of the complex (Fig. 2e). The PZ-2891 interactions are described with residues on the opposite protomer indicated with a prime (Fig. 2f, g). The isopropyl moiety packs into a hydrophobic cavity created by V250′, I253′, Y254′, Y258′, and A269′ located on the flexible flap that is disordered in the PANK3•AMPPNP•Mg$^{2+}$ complex[9] but becomes structured in the PANK3•AMPPNP•Mg$^{2+}$•PZ-2891 complex (Fig. 2f, g). The carbonyl group forms hydrogen bond interactions with R207, which explains the lack of PZ-3067 binding (Fig. 1c). The piperazine ring acts as a spacer to present the pyridazine ring to the opposite protomer to hydrogen bond with R306′ and engage W341′ by π-π stacking interactions. PZ-2891 interaction with R306′ also allows an inter-protomer hydrogen bond between R306′ and T209 to further stabilize the dimer. Comparison of the PANK3•AMPPNP•Mg$^{2+}$•PZ-2891 complex to the PANK3•AMPPNP•Mg$^{2+}$•pantothenate complex

(Fig. 2g) shows that pantothenate and PZ-2891 both form hydrogen bonds with R207, and the isopropyl substituent of PZ-2891 docks into the hydrophobic cavity occupied by the dimethyl group of pantothenate. However, pantothenate does not interact with the dimer interface, and the connection between R306′ and T209 in the pantothenate complex is mediated by a water molecule rather than by a direct hydrogen bond. The structural analysis suggests that PZ-2891 first binds to the pantothenate site of the PANK3•ATP•Mg$^{2+}$ complex followed by closing of the flexible loop, and the engagement and re-organization of the dimer interface. The induced fit mechanism is the most common binding mode observed in high-affinity inhibitors[37], and with PZ-2891 the two protomers become locked together to create a structure that is distinct from the structures of all the normal catalytic intermediates[9].

**Pantazines are allosteric PANK activators.** The idea behind the deployment of pantazines as PANK activators in cells was to exploit the high cooperativity of the enzyme to lock it in an active conformation that cannot be inactivated by acetyl(acyl)-CoA. This unusual effect was predicted from the established interactions between PANK, its substrates, acetyl-CoA inhibitor and pantazine outlined in Fig. 3a. At subsaturating pantazine concentrations, a new drug-induced catalytic cycle exists that prevents PANK3 from returning to a conformation that can bind to, and be inhibited by, acetyl-CoA. Within the cell, the PANK3 dimer exists in one of two distinct conformations: the inactive conformation stabilized by the binding of acetyl-CoA and the active conformation stabilized by the binding of ATP•Mg$^{2+}$. In the normal catalytic cycle, ATP cooperatively binds to the PANK dimer switching both protomers to the active, closed conformation. Pantothenate binds and catalysis proceeds through the three, structurally characterized intermediates, and both of the active sites empty[9]. The phosphopantothenate product is rapidly converted to CoA by the biosynthetic pathway. Acetyl-CoA is a feedback inhibitor of PANK, and cellular acetyl-CoA levels rise until almost all of the PANK exists in the acetyl-CoA-bound, inactive protein conformation.

Sub-saturating PZ-2891 concentrations interrupt this normal catalytic cycle and feedback regulatory mechanism. In the in vitro biochemical assays to optimize the pantazine series, acetyl-CoA was absent and the presence of ATP (>1 mM) ensured that PANK3 existed only in the active conformation. Under these conditions, PZ-2891 only acts as an inhibitor by titrating the active sites (Fig. 3a). However, the presence of acetyl-CoA in cells means that upon completion of the catalytic cycle PANK3 is available to bind acetyl-CoA and switch to the inactive conformation. When PZ-2891 is bound to only one protomer, the opposite protomer remains capable of catalysis. However, at the end of the pantazine-dependent catalytic cycle, PANK3 remains locked in its active conformation by the pantazine preventing binding of the acetyl-CoA inhibitor and allowing another cycle of catalysis to proceed (Fig. 3a). This effect renders PANK3 dimers with PZ-2891 bound to only one protomer refractory to feedback inhibition by acetyl-CoA (Fig. 3a).

The unusual effect of PZ-2891 acting as both an orthosteric inhibitor and an allosteric activator of PANK3 activity in the presence of acetyl-CoA was tested in biochemical assays designed to mimic the mixture of ligands present in cells. The assays contained either PANK3 or PANK3 plus PZ-2891 at a concentration of drug that partially inhibited (25%) the total PANK3 activity. In the absence of PZ-2891, PANK3 activity was extinguished in a concentration-dependent manner by acetyl-CoA as is normally observed[6] (Fig. 3b). However, in the presence of PZ-2891 approximately half of the PANK3 activity that

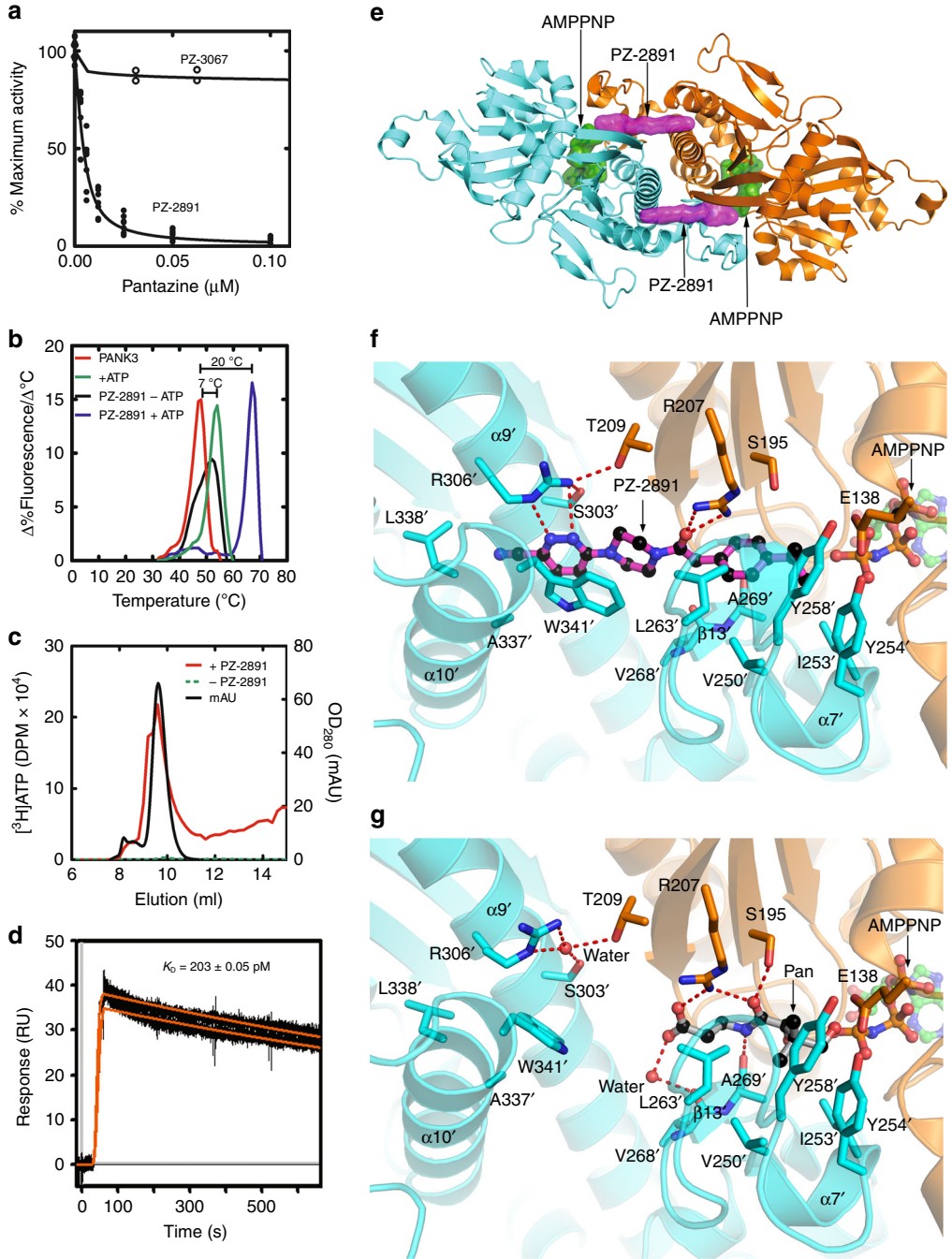

**Fig. 2** PZ-2891 binds to the PANK3•ATP•Mg$^{2+}$ complex with nM affinity. **a** PZ-2891 ($n = 10$) (filled circles) and PZ-3067 (open circles) ($n = 2$) inhibition of PANK3. The data were fit to the Morrison equation and $1.3 \pm 0.2$ nM was the consensus PZ-2891 IC$_{50}$ from five experiments using different batches of purified enzyme. A representative data set is shown. **b** Thermal stabilization of PANK3 by 8 mM ATP, 8 μM PZ-2891 or 2 mM ATP plus 2 μM PZ-2891. Control (red line), PZ-2891 (black line), ATP (green line), and ATP plus PZ-2891 (blue line). The peaks of the first derivative plots of the thermal denaturation curves identify the temperatures at which 50% of the protein is unfolded. The shifts in the thermal denaturation temperatures were calculated from experiments performed in triplicate, and the averages rounded to the nearest degree. **c** Isolation of the PANK3•[$^{3}$H]ATP•Mg$^{2+}$•PZ-2891 complex by gel filtration chromatography. Black trace, PANK3 protein elution profile (A$_{280}$); Red trace, [$^{3}$H]ATP elution profile in the presence of PANK3 + PZ-2891; Green trace (on baseline), [$^{3}$H]ATP elution profile with PANK3 alone. One example of duplicate experiments is shown. **d** Surface plasmon resonance analysis of PZ-2891 binding to PANK3 in the presence of 1 mM ATP•Mg$^{2+}$. The data (black lines) were fit to a 1:1 binding model (orange lines). The association ($k_a$), dissociation ($k_d$), equilibrium ($K_D$) constants were calculated as $2.37 \times 10^{6}$ M$^{-1}$s$^{-1}$, $4.82 \times 10^{-4}$ s$^{-1}$ and 0.203 nM respectively. The residence time ($1/k_d$) of PZ-2891 was 34 min. Data are the average of two experiments in duplicate. **e** Overview of the PANK3 dimer illustrating that PZ-2891 binds across the dimer interface. The two PANK3 protomers are colored cyan and gold, PZ-2891 is purple and AMPPNP is green. **f** Close-up view of PZ-2891 bound across the dimer interface of the PANK3•AMPPNP•Mg$^{2+}$ complex illustrating the key hydrogen bonding interactions (dotted red lines) with both PANK3 protomers. **g** Structure of the PANK3•AMPPNP•Mg$^{2+}$•pantothenate (Pan) complex (PDB ID: 5KPR) with the pantothenate hydroxyl rotated into the catalytically active position

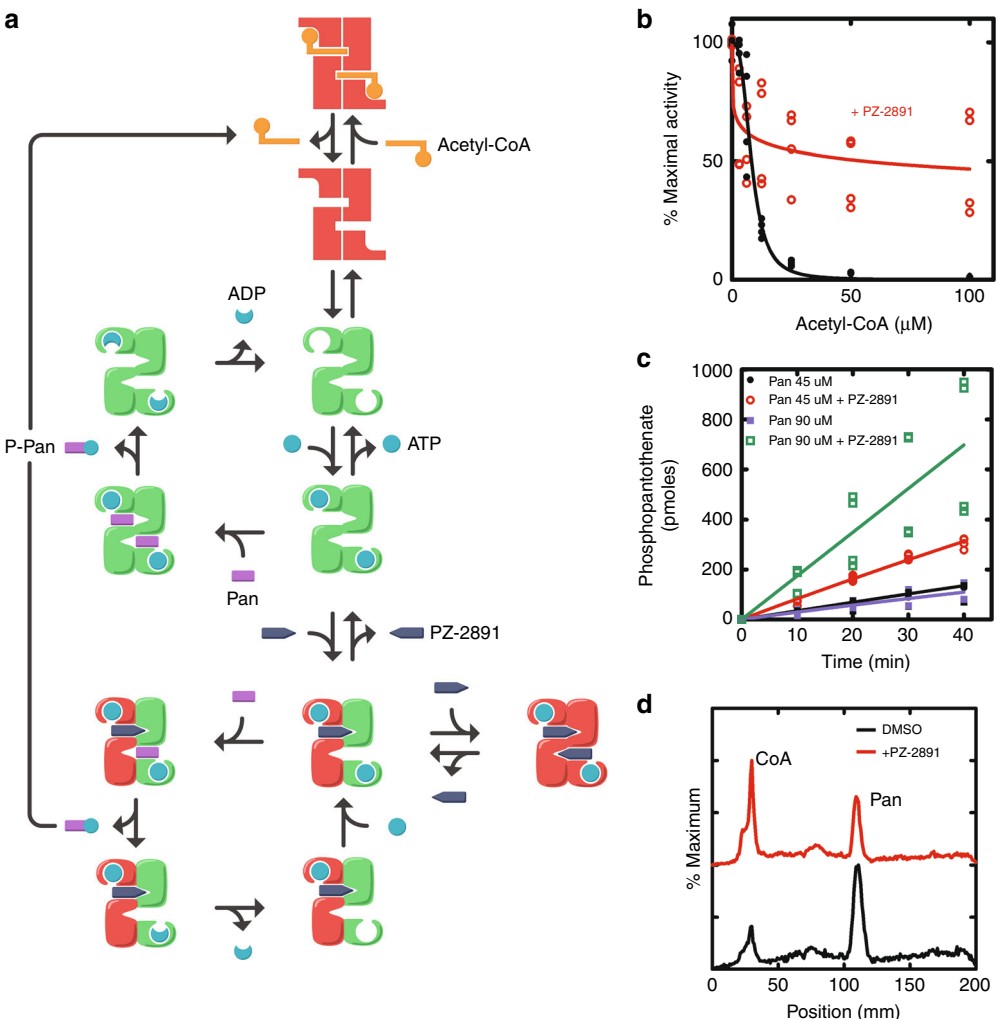

**Fig. 3** Activation of PanK by PZ-2891. **a** Kinetic model illustrating how PZ-2891 binding activates PANK3. PANK3 is a highly-cooperative enzyme where both active sites exist in either the active or inactive conformation. The inactive 'open' conformation (red rectangles) is stabilized by the binding of acetyl-CoA, and the active 'closed' conformation (green or curved shape) is stabilized by ATP binding. In the normal catalytic cycle, PANK3 returns to a ligand-free state that allows it to bind acetyl-CoA and switch to the inactive conformation. PZ-2891 binds tightly to the active, ATP-bound PANK3 conformation. When a fraction of PANK3 is occupied by PZ-2891, the catalytic cycle is operating on only one protomer of the dimer. In this catalytic cycle, the active monomer of PANK3 empties, but remains locked in the active conformation by ATP•Mg$^{2+}$•PZ-2891 binding to the other protomer. The drug-induced catalytic cycle prevents PANK3 from adopting the open, inactive conformation making PANK3 refractory to feedback regulation by acetyl-CoA. Pan pantothenate, P-Pan phosphopantothenate. **b** PZ-2891 renders PANK3 refractory to acetyl-CoA inhibition. PANK3 (1 µg/assay) activity at different concentrations of acetyl-CoA was determined with ATP (1 mM) in the presence (open red circles) and absence (filled circles) of 2.5 µM PZ-2891. **c** The time course assays contained PANK3 (1 µg/assay), ATP (1 mM) and acetyl-CoA (100 µM) such that PANK3 was inhibited about 95% by acetyl-CoA in the absence of pantazine. The addition of 2.5 µM PZ-2891 to the assay increased the PANK3 rate in a pantothenate-dependent manner. Filled black circles are 45 µM pantothenate, filled blue squares are 90 µM pantothenate, red open circles are 45 µM pantothenate plus 2.5 µM PZ-2891, and green open squares are 90 µM pantothenate plus 2.5 µM PZ-2891. Data in **b** and **c** are from two independent experiments that were performed in duplicate. **d** C3A cells were radiolabeled with [$^3$H]pantothenate in the presence (red trace) or absence (black trace) of 10 µM PZ-2891 for 24 h. The cells were extracted, and the labeled metabolites were analyzed by thin-layer chromatography and imaged with a Bioscan detector. Representative chromatograms from two independent experiments in duplicate are shown

remained in the presence of PZ-2891 was refractory to acetyl-CoA inhibition (Fig. 3b). These data directly support the existence of the drug-induced catalytic cycle that prevents the enzyme from interacting with acetyl-CoA. A second experiment to model the cellular environment employed PANK3 whose activity was repressed >95% by the presence of acetyl-CoA (Fig. 3c). Under these conditions, an increase in pantothenate from 45 µM to 90 µM had little impact on the rate because PANK3 existed in a conformation unable to bind the substrate. The addition of PZ-2891 activated PANK3, and increasing the

pantothenate concentration to twice its Km further accelerated PANK3 activity (Fig. 3c). These data are consistent with pantazines not affecting the pantothenate Km (Supplementary Fig. 2c), and predict that pantothenate concentrations above the Km for PANK would provide the highest level of pantazine-dependent PANK activity in the presence of acetyl-CoA. These data predict that pantazine-treated cells would have elevated CoA and reduced pantothenate due to the activation of PANK. C3A cells were labeled with [$^3$H]pantothenate in the presence and absence of 10 µM PZ-2891 and the CoA and pantothenate levels

were determined following separation by thin-layer chromatography (Fig. 3d). PZ-2891 treatment increased the amount of intracellular CoA and lowered intracellular pantothenate, thus illustrating the operation of the pantazine-dependent catalytic cycle that functions to elevate CoA in cells (Fig. 3a).

**PZ-2891 increases intracellular CoA in cultured cells.** As a prelude to examining the pantazine effect in animals, a series of experiments were performed to verify that PZ-2891 targeted PANK3 in cells and to determine if there were any obvious off-target contraindications. A cellular thermal shift assay (CETSA)[38] was used to confirm that PZ-2891 was bound to cellular PANK3. HEK 293 T cells transfected with a plasmid expressing PANK3 were used to increase cellular levels of PANK3 so that the protein could be detected by immunoblotting cell lysates (Fig. 4a). The CETSA assay showed that PZ-2891 stabilized PANK3 by ~7 °C confirming that PZ-2891 bound to PANK3 in intact cells (Fig. 4b). A PZ-2891 CETSA dose-response curve indicated that half of the cellular PANK3 was bound at a concentration of ~9 μM PZ-2891 (Supplementary Fig. 5). PZ-2891 (10 μM) did not impact cell viability as measured by cell growth or the synthesis of DNA, protein and lipid, indicating the absence of significant toxicity toward cultured cells (Supplementary Fig. 6). The propensity of PZ-2891 to interfere with the activity of other cellular processes was evaluated by two enzymatic screens. First, PZ-2891 did not have significant inhibitory activity in a screen of 468 mammalian kinases (Supplementary Fig. 7, Supplementary Dataset 1). The second screen examined the effect of PZ-2891 on 72 cellular proteins known to cause off-target effects in drug discovery (Supplementary Dataset 2). These data show that PZ-2891 selectively targeted PANK in cells and indicated the absence of off-target pharmacological interactions.

The total intracellular CoA levels were determined in a human liver-derived cell line (C3A) treated for 24 h with increasing concentrations of PZ-2891 (Fig. 4c). CoA progressively increased up to 10 μM PZ-2891, whereas 20 and 50 μM were less activating. As expected, cells did not respond to the inactive pantazine, PZ-3067 (Fig. 4c). The activation phenomenon was not due to enhanced transcription of the kinase isoforms (Supplementary Fig. 8). To verify that PZ-2891 action was PANK-dependent, cells were transfected with an expression plasmid to increase the cellular content of PANK3. The addition of 10 μM PZ-2891 to control (empty vector) HEK293T cells also significantly increased the levels of intracellular CoA (Fig. 4d). The plasmid-driven increase in PANK3 expression resulted in higher baseline CoA, and the treatment of PANK3 overexpressing cells with 10 μM PZ-2891 triggered a large increase in intracellular CoA (Fig. 4d). Again, PZ-3067 was inactive. Similar transfection experiments showed that PANK1 expression also raised CoA after a 24 h treatment with 10 μM PZ-2891 (Fig. 4e). There was no PZ-2891-stimulated CoA synthesis in cells expressing the inactive PANK3 (E138A) mutant, consistent with a requirement for catalytically active PANK3 rather than some other function of the protein[9] (Fig. 4e). Along with CoA, the levels of phosphopantetheine and dephospho-CoA and other pathway intermediates were measured (Supplementary Fig. 9). These two pathway intermediates were minor components in both control and PZ-2891-treated cells showing that activation of CoA biosynthesis at the PANK step was not constrained by a secondary control point in the CoA biosynthetic pathway. Mass spectrometry showed that PZ-2891-treated cells had higher acetyl-CoA levels compared to untreated controls (Supplementary Fig. 10), indicating that PZ-2891 effectively prevented feedback inhibition by acetyl-CoA. Once PANK3 is in the active state, the reaction rate is determined by the concentration of pantothenate in relation to the PANK Km

(Fig. 3c). A reduction in the pantothenate in the cell culture medium reduced PZ-2891 activation of CoA synthesis, whereas increasing the media concentration potentiated the PZ-2891 effect (Fig. 5a). Pantothenate itself had no effect on CoA content, consistent with PANK as the feedback-regulated, rate-controlling step in the pathway.

**PZ-2891 elevates tissue CoA content in mice.** The favorable in vitro ADME and in vivo pharmacokinetic properties of PZ-2891 (Supplementary Tables 1, 2) suggested it could be utilized as a proof of principle pantazine to alter CoA levels in a mouse model. Mice were treated with PZ-2891 either with or without a pantothenate supplement included in the oral gavage to determine if pantothenate administration augmented the effect of PZ-2891 in mouse liver as it did in cultured cells. Treatment of mice with the maximum soluble dose of PZ-2891 (30 mg/kg) increased total liver CoA by 50% in animals administered PZ-2891 alone. There was a 100% increase in CoA levels in animals treated with PZ-2891 plus pantothenate (Fig. 5b). We next analyzed target tissue pantothenate levels to determine the pantothenate concentrations in animals maintained on normal chow (25 ppm pantothenate) or animals treated with 200 mg/kg of pantothenate by oral gavage. Pantothenate supplementation increased pantothenate levels in plasma (Fig. 5c), liver (Fig. 5d), forebrain (Fig. 5e) and hindbrain (Fig. 5f). The amounts of pantothenate in liver correlated with the plasma levels, and calculation of the intracellular concentrations in liver showed higher levels of pantothenate than in the plasma in both cases (10.6, and 24.4 μM, respectively). Similarly, pantothenate levels in the hindbrain and forebrain were elevated by pantothenate supplementation; however, the pantothenate levels in brain were significantly higher than in liver. In all circumstances, estimated tissue pantothenate concentrations were below the PANK3 pantothenate Km (45 μM), explaining how the pantothenate supplement potentiates the effect of PZ-2891 on tissue CoA levels.

The potential therapeutic benefit of a treatment for PKAN neurodegeneration rests on its ability to cross the blood–brain barrier and increase CoA levels in the brain. Male and female mice were co-administered increasing PZ-2891 concentrations to determine if PZ-2891 was capable of elevating CoA in the brain (Fig. 6). The animals received 5 doses of PZ-2891 by oral gavage every 12 h, and tissues were harvested 4 h after the last dose. We observed a dose-dependent increase in total CoA in liver (Fig. 6a, d), forebrain (Fig. 6b, e) and hindbrain (Fig. 6c, f) in PZ-2891-treated male and female animals. PZ-2891 increased CoA in both males and females to a similar extent. In this short treatment regimen, higher PZ-2891 doses were required to maximally increase CoA in the brain compared to the liver. We also measured PZ-2891 levels in tissues from mice treated with PZ-2891 for four weeks, and these data clearly show the presence of drug in liver and brain (Supplementary Fig. 11). To ensure that an active PANK2 was not necessary for the PZ-2891 effect, we treated $Pank2^{-/-}$ mice with PZ-2891 and measured the CoA levels in liver and brain. PZ-2891 effectively increased CoA in the tissues of $Pank2^{-/-}$ knockout mice (Supplementary Fig. 12) illustrating that the expression of PanK2 was not required for pantazine activation of CoA biosynthesis, and that activation of PanK1 and PanK3 were sufficient to raise CoA. PZ-2891 therapy would be maintained for extended periods, therefore we formulated pantothenate-supplemented mouse chow with different concentrations of PZ-2891. Five mice were treated for 1 week with 112 ppm PZ-2891 in the chow, and the elevation in CoA in six tissues were examined (Supplementary Fig. 13). CoA levels were elevated in liver, brain, and to a lesser extent, heart. Muscle CoA levels are the lowest of all tissues[39], and were not elevated by

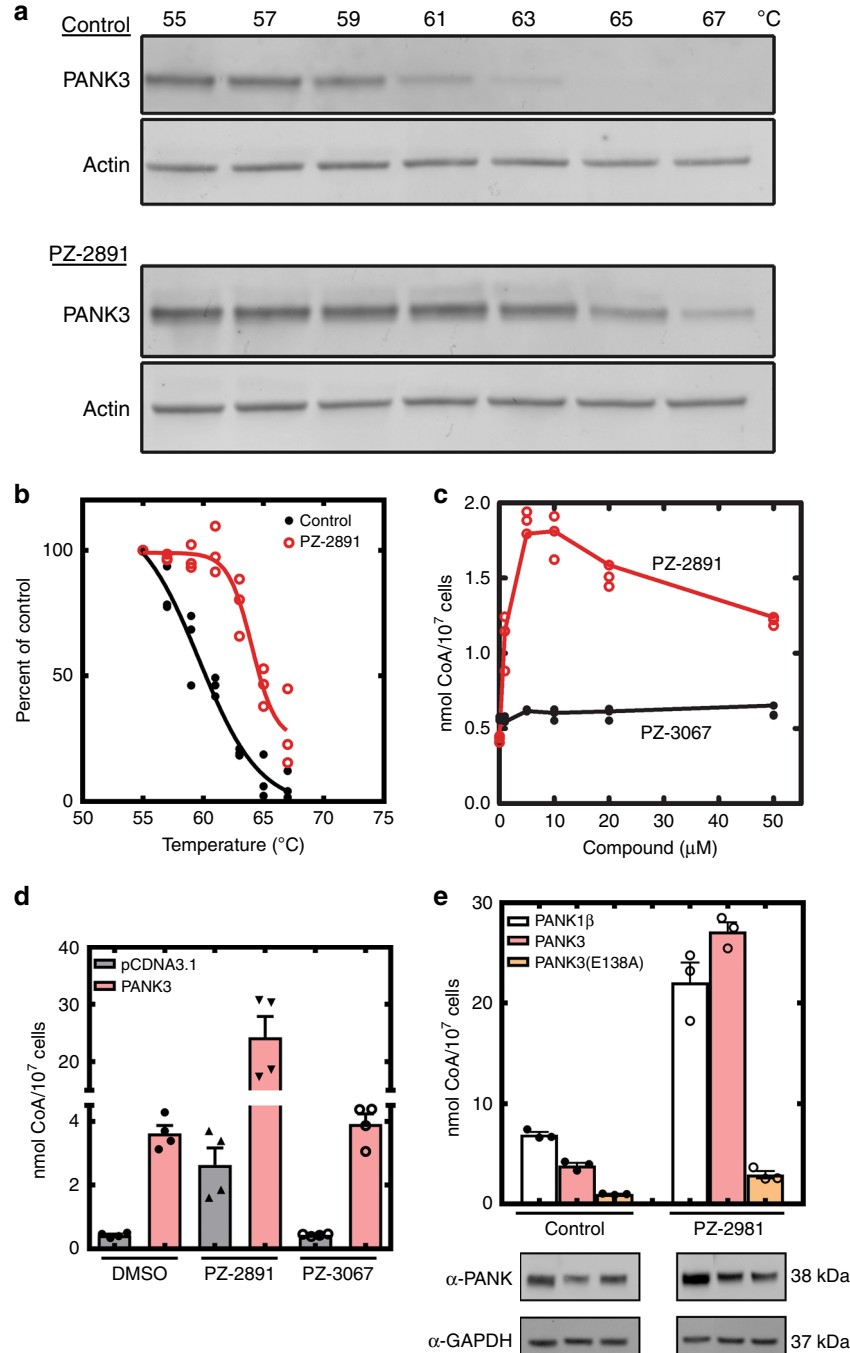

**Fig. 4** Pantazine modulation of intracellular CoA levels. **a** Representative immunoblots from a cellular thermal shift (CETSA) assay showing the stabilization of PANK3 by PZ-2891. **b** PANK3 stability in the presence (red open circles) and absence (filled black circles) of 10 µM PZ-2891. These data were obtained from three independent biological experiments. **c** Intracellular CoA levels in C3A cells. Cell cultures were treated with either 10 µM PZ-2891 (red open circles) or its inactive homolog, PZ-3067 (filled black circles), and the CoA levels were determined. These data were derived from three independent biological experiments. **d** HEK293T cells were transfected with either an empty vector (gray bars) or vector driving the expression of PANK3 (pink bars), and the cells were treated with either 10 µM PZ-2891 or PZ-3067. The data are from two independent biological experiments performed in duplicate and the mean ± SEM is plotted. **e** HEK293T cells were transfected with expression vectors for either PANK1β (white bars), PANK3 (pink bars), or PANK3 (E138A) (orange bars) . PANK3(E138A) is a catalytically inactive mutant. Triplicate dishes of cells were treated for 24 h with either DMSO or 10 µM PZ-2891 in DMSO, the cells were harvested and the total intracellular CoA levels determined. Data were plotted as the mean ± SEM. Isoform expression was verified by immunoblotting cells samples with an anti-PANK antibody (α-PANK) raised against the catalytic core common to all isoforms

PZ-2891 therapy. A dose-response experiment was conducted by the administration of a series of PZ-2891 doses in the diet (Fig. 6g, i). These experiments showed both liver and brain CoA were significantly elevated by PZ-2891 in a dose-dependent manner. During the one-month treatment there was no difference in the weights or behavior of the treated animals. These data established that long-term PZ-2891 therapy was effective in elevating CoA in the liver and brain of mice.

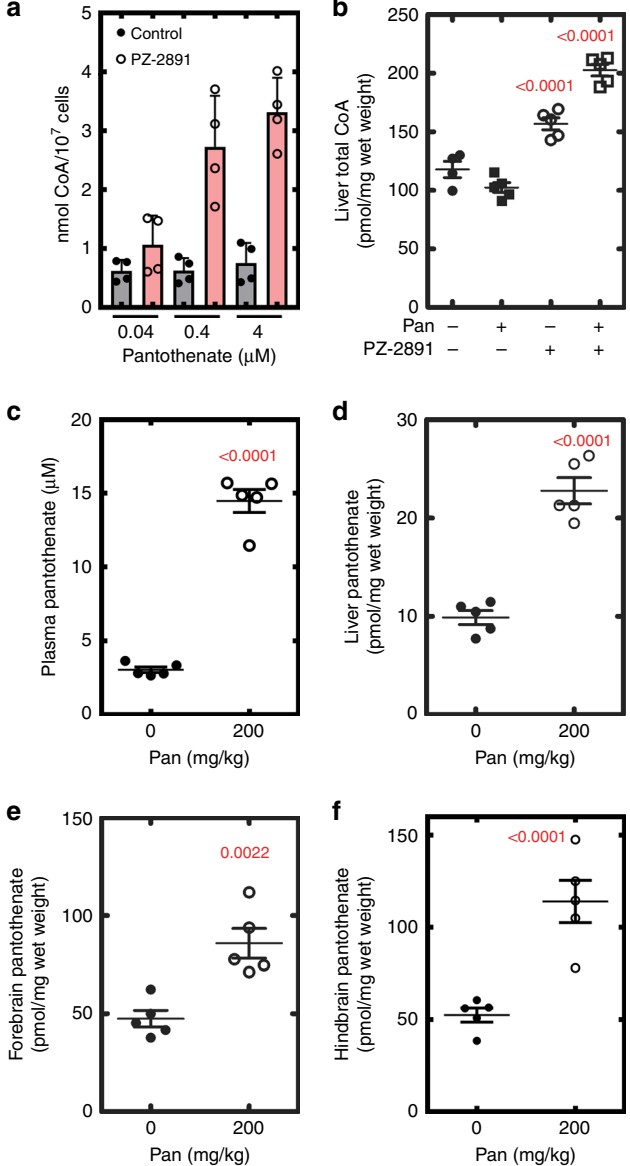

**Fig. 5** Pantothenate concentration and PZ-2891 activation of CoA synthesis in animals. **a** Pantothenate-dependent PZ-2891 elevation of CoA in cultured C3A cells. Pantothenate levels were manipulated by supplementing pantothenate-free DMEM + dialyzed FBS. Control CoA levels (gray bars) are compared to cells treated with 10 μM PZ-2891 (pink bars). Data are from two experiments performed in duplicate, and plotted as the mean ± SEM. **b** Groups of 5 mice were treated with five doses of PZ-2891 (30 mg/ kg) without or with 200 mg/kg pantothenate. PZ-2891 was administered at 12 h intervals, and 4 h after the last dose the animals were euthanized and the CoA content of the tissues determined. Untreated mice (filled circles); pantothenate-treated mice (filled squares), PZ-2891 treatement (open circles), and PZ-2891 + pantothenate (open squares). **c–f** Groups of 5 animals were gavaged with 200 mg/kg of pantothenate using the same regimen as in **b**, and four hours after the last dose, tissues were harvested and the pantothenate concentrations measured by mass spectrometry. **c** Plasma. **d** Liver. **e** Forebrain. **f** Hindbrain. Mean ± SEM are plotted, and the p values are in red. Control pantothenate levels (filled circles), pantothenate levels in pantothenate-treated mice (open circles)

## PZ-2891 therapy increases lifespan in a mouse model of brain CoA deficiency

We developed a mouse model for neuronal CoA deficiency as a platform to test therapeutics (Fig. 7). These animals have both the *Pank1* and *Pank2* genes as homozygous 'floxed' alleles, and the Cre recombinase controlled by the synapsin I promoter (*SynCre*) transgene gives rise to neuron-selective deletion of both alleles (Supplementary Fig. 14). Control animals were littermates lacking the *SynCre* transgene. The brains of *SynCre* + animals analyzed by RT-PCR showed the reduction in both *Pank1* and *Pank2* mRNA, but not *Pank3*, in brain (Supplementary Fig. 15a). The same analysis on liver tissue showed that all three *Pank* genes were normally expressed, illustrating the selective deletion of *Pank1/Pank2* in the brain (Supplementary Fig. 15b). The mice develop normally until postnatal day 12.5, at which time they begin to lose weight (Fig. 7a), and exhibit a median survival of 52 days (Fig. 7b). PZ-2891 (112 ppm in chow) treatment began at weaning (day 21) after the disease was established and significant weight loss had occurred. The treated animals immediately began to gain weight after therapy was initiated (Fig. 7a), and had a median survival of 150 days (Fig. 7b). Two of 11 treated animals were so affected that they died within two days of starting PZ-2891, and 5 of 11 treated animals lived until the study was terminated at six months. Day 45 was selected as a time to compare treated and untreated animals (Fig. 7a). The *SynCre*+ animals had lower CoA in the brain compared to their littermate controls, and PZ-2891 therapy significantly elevated CoA in both the forebrain and hindbrain (Fig. 7c, d). The *SynCre*+ animals exhibited a significant locomotor defect as measured in the open field test, an accepted method for assessing mouse neuromuscular disorders. The movement of *SynCre*+ animals was severely impaired (Fig. 7e), and when they did move, they did not travel far (Fig. 7f). PZ-2891 treatment of the *SynCre*+ animals significantly increased locomotor activity (Fig. 7e,f; Supplementary Fig. 16). PZ-2891 did not affect control mouse locomotor activity. These data show that PZ-2891 therapy increases weight, life span, and brain CoA levels, as well as improving the locomotor activity of animals with severe growth and locomotor defects due to a deficiency of pantothenate kinase activity and CoA in neurons.

## Discussion

PKAN is a progressive neurodegenerative disorder affecting movement, balance, speech, vision, cognition and behavior, and it arises from debilitating mutations in the PANK2 gene[4]. PANK2 is a major PANK isoform expressed in the brain, and PKAN symptoms are thought to arise from a CoA deficiency that compromises important neuronal processes including iron metabolism, synaptic transmission, synaptic vesicle cycling, neuron projection development, and protein quality control[40]. There is no effective PKAN treatment. Our therapeutic approach is to compensate for the loss of PANK2 by activating the two other PANK isoforms using a small molecule (PZ-2891) that penetrates the blood brain barrier to elevate tissue CoA. PZ-2891 is the result of a successful selection/ optimization process designed to exploit the high cooperativity between PANK protomers that simultaneously switch from an active ATP-bound state to an inactive acetyl-CoA-bound state. PZ-2891 exploits this mechanism by engaging PANK across the dimer interface to act as an orthosteric inhibitor at high concentrations and an allosteric activator at lower sub-saturating concentrations. Thus, PZ-2891 relaxes the feedback inhibition of PANK resulting in a significant rise in CoA levels in cells and tissues. Due to the neuronal nature of PKAN, the design process incorporated physicochemical properties that reflect the top 25 drugs which target the central nervous system[31,32], resulting in blood brain barrier penetration and the elevation of brain CoA levels. A mouse model of neuronal CoA deficiency was developed to evaluate the therapeutic potential of the pantazines. These animals have reduced CoA in the brain, fail to gain weight, have a severe locomotion defect, and a reduced life span. PZ-2891 treatment elevates brain CoA, the

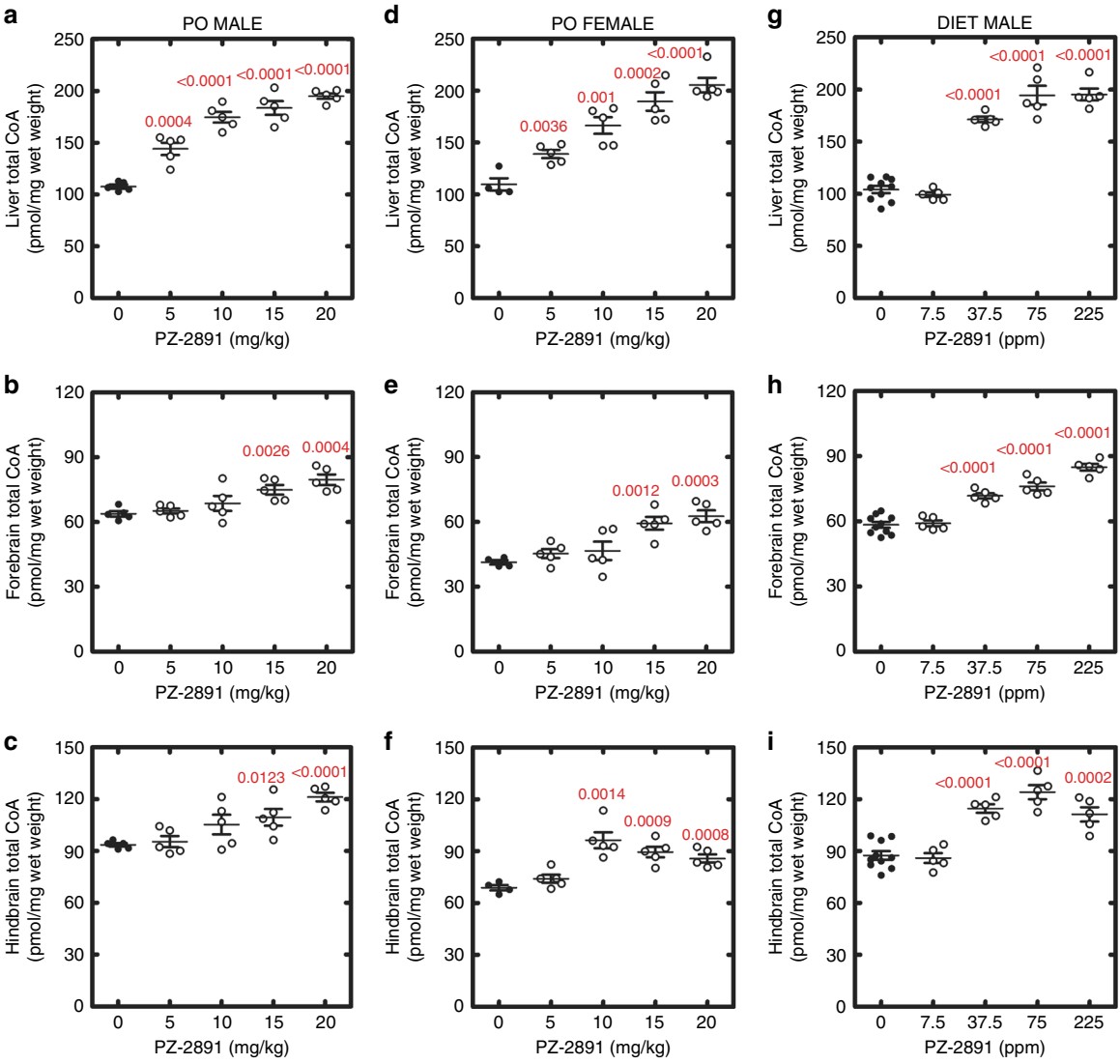

**Fig. 6** Activation of CoA synthesis in liver, forebrain and hindbrain of male and female animals treated with PZ-2891. Groups of 5 mice were administered the indicated amounts of PZ-2891 plus 100 mg/kg pantothenate by oral gavage every 12 h for 5 doses. Four hours after the last dose, the tissues were harvested and the tissue CoA levels determined. Control CoA levels (closed circles); PZ-2891 treated CoA levels (open circles). **a** Male liver. **b** Male forebrain. **c** Male hindbrain. **d** Female liver. **e** Female forebrain. **f** Female hindbrain. Groups of 5 mice were maintained on chow fortified with 1000 ppm pantothenate and the indicated levels of PZ-2891. The mice were maintained on the diets for 4 weeks and their tissues were harvested and the CoA levels determined. **g** Liver CoA levels. **h** Forebrain CoA levels. **i** Hindbrain CoA levels. Statistical significance was determined using Student's t-test calculated with Graph-Pad software and the p values (red) are noted on the figure panels in red. Means ± SEM are plotted

treated animals gain weight, their lifespan increases significantly, and they exhibit increased locomotor activity. These data support the conclusion that the activation of alternate pantothenate kinase isoforms by pantazines is a promising approach to PKAN therapy.

The pantazines are also powerful chemical probes to interrogate the role of CoA homeostasis in cell physiology. The biochemical mechanism of activation of PZ-2891 is different from other previously described allosteric kinase activators that act by either destabilizing the inactive kinase conformation or stabilizing the active one[41,42]. This unique mechanism of action makes PZ-2891 a highly specific molecule that does not significantly interact with other mammalian kinases or receptors. Another attractive feature of PZ-2891 is its relatively long residence time on PANK. Residence time on the cellular target is a key parameter in governing drug efficacy[37], and in the case of PZ-2891, mitigates its relatively fast metabolic clearance from the body. Our studies on

the cellular effects of PZ-2891 verify that pantothenate kinase is the only relevant rate-controlling enzyme in mammalian CoA biosynthesis. Although CoA synthase was suggested as a second control point in the CoA biosynthetic pathway[6,43–45], our quantitative identification of pathway intermediates by mass spectrometry shows that significant activation of PANK does not result in the accumulation of other pathway intermediates, such as phosphopantetheine, the CoA synthase substrate. The steady state CoA tissue level is controlled by the relative rates of synthesis and turnover. CoA turnover is thought to be catalyzed by nudix hydrolases[11,46], but in cultured cells, nudix hydrolase expression is low and CoA turnover may be slower than in animal tissues[47]. Current medicinal chemistry efforts are directed toward optimizing the pharmaceutical properties of PZ-2891 while maintaining its excellent oral bioavailability, target affinity, and blood brain barrier permeability.

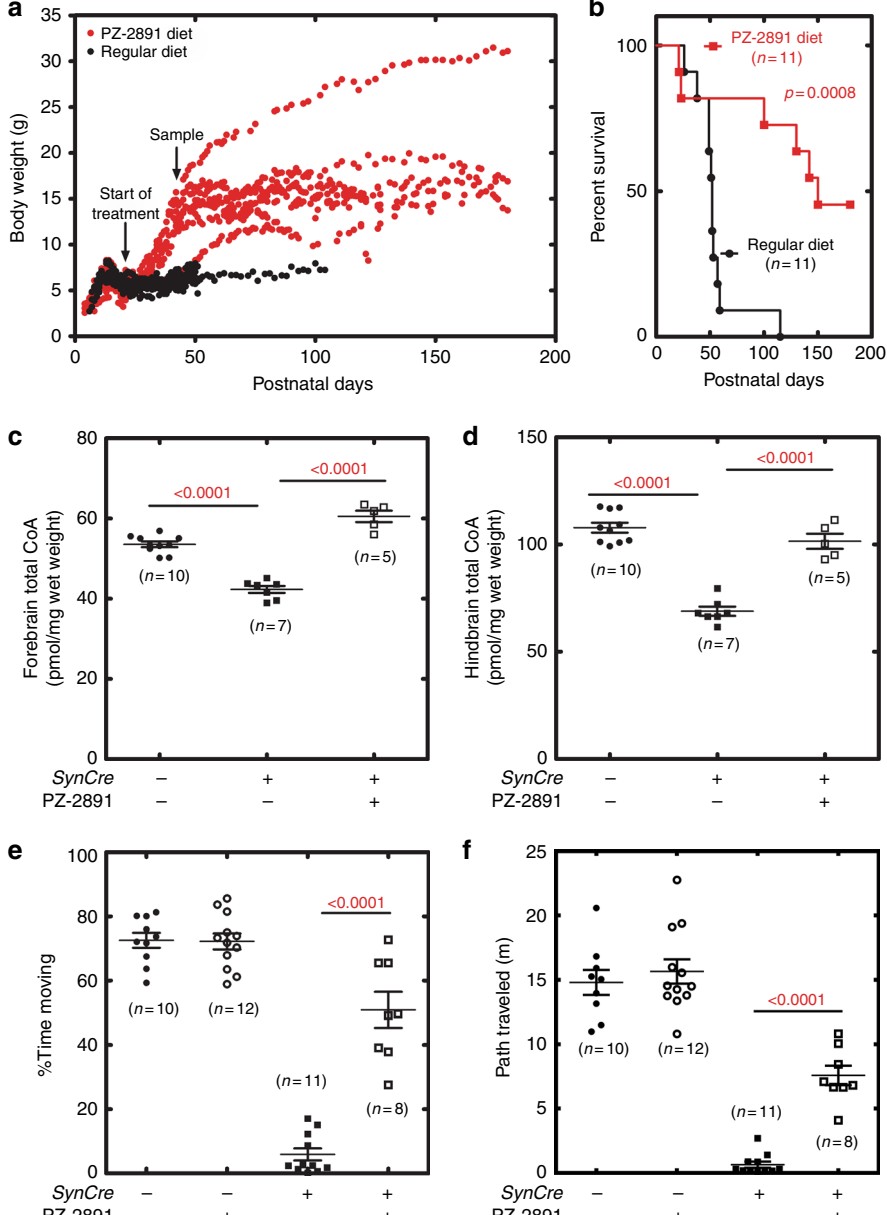

**Fig. 7** PZ-2891 therapy in *SynCre+ PANK1,PANK2* neuronal knockout mice. Male and female mice were randomized into the experimental groups when they were genotyped, and the numbers of mice used in each analysis are shown in parenthesis in the figure panels. **a** Weight monitoring of double knockout mice maintained on the control diet (black circles) or on a diet with PZ-2891 (red circles). The start of treatment and the time of sample collection are indicated with arrows on the figure. **b** Lifespan of double knockout mice with (red line) or without (black line) PZ-2891 treatment. **c** Forebrain CoA levels in littermate control (*SynCre−*) mice (filled circles) compared to *SynCre+ Pank1,Pank2* neuronal knockout mice at day 45 with (open squares) or without (filled squares) PZ-2891 therapy. **d** Hindbrain CoA levels in littermate control (*SynCre−*) mice (filled circles) compared to *SynCre+ Pank1,Pank2* neuronal knockout mice at day 45 with (open squares) or without (filled squares) PZ-2891 therapy. **e** The percent time mice were moving during the 5 min open field test. Littermate control (*SynCre−*) mice (untreated, filled circles; treated open circles) were compared to *SynCre+ Pank1,Pank2* neuronal knockout mice (untreated filled squares; treated, open squares) at day 45 with or without PZ-2891 therapy. **f** Distance traveled by mice in the 5 min open field test. Littermate control (*SynCre−*) mice (untreated, filled circles; treated, open circles) were compared to *SynCre+ Pank1,Pank2* neuronal knockout mice at day 45 with (open squares) or without (closed squares) PZ-2891 therapy. The *p* values are in red, and means ± SEM are plotted

## Methods

**Materials.** D-[1-$^{14}$C]Pantothenate (specific activity, 55 mCi/mmol) from American Radiolabeled Chemicals; Ni-NTA resin from Qiagen; [$^{3}$H]ATP from Perkin Elmer (specific activity, 29.8 Ci/mmol); Sypro Orange dye from Thermo Fisher Scientific; D-[$^{3}$H]pantothenate (specific activity, 50 Ci/mmol) from American Radiolabeled Chemicals; D-pantothenic acid, hemicalcium salt from Sigma-Aldrich; pantothenate-free DMEM from Thermo Fisher Scientific; CoA thioesters from Avanti Polar Lipids. All other materials were reagent grade or better.

**Dynamic-injection surface plasmon resonance**. Experiments were conducted at 20 °C using a SensiQ Pioneer optical biosensor (SensiQ Technologies). His-tagged human PANK3 was immobilized on polycarboxylate hydrogel-coated gold chips preimmobilized with nitrilotriacetic acid (HisCap chips; SensiQ Technologies). The chip was primed in chelating buffer (10 mM HEPES, pH 7.4, 150 mM NaCl, 50 μM EDTA, 0.005% Tween-20) and was preconditioned at 10 μl/min with three 60 s injections of wash buffer (10 mM HEPES, pH 8.3, 150 mM NaCl, 350 mM EDTA, 0.05% Tween-20) and one 60 s injection of chelating buffer before being charged with a 60 s injection of 500 μM NiCl$_2$ in chelating buffer. After priming into

binding buffer (20 mM Tris-HCl, pH 7.5, 200 mM NaCl, 1 mM TCEP, 2 mM MgCl$_2$, 1 mM ATP, 0.005% Tween-20, 3% DMSO), PANK3 was injected until ~ 3000–4000 RU of protein were captured. The reference flow cell on the chip was charged with Ni$^{2+}$ without adding protein.

PZ-2891 was prepared in binding buffer, and a gradient of 0–50 or 0–100 nM was injected in duplicate for each concentration at a flow rate of 200 μl/min using the OneStep Injection feature, which exploits Taylor dispersion to generate a concentration gradient that provides a full titration of analyte in a single injection. A series of buffer-only (blank) injections was included to account for instrument noise. PANK3 was released from the chip with 500 mM imidazole and recaptured for each cycle due to the slow dissociation of the compound. The data were processed, double-referenced, solvent corrected and analyzed using the software package Qdat (version 2.5.3.5, SensiQ Technologies). Kinetic rate constants ($k_a$, $k_d$) were determined by globally fitting the data at all concentrations to a 1:1 (Langmuir) binding model. Equilibrium dissociation constants were calculated as the quotient $K_D = k_d/k_a$.

**PANK activity assays.** PANK activity assay was performed in the presence of 0–10 μM compound in a reaction mixture that contained 100 mM Tris-HCl, pH 7.5, 10 mM MgCl$_2$, 2.5 mM ATP, 45 μM D-[1-$^{14}$C]pantothenate (specific activity, 22.5 mCi/mmol) and 5 nM of human PANK3. PANK3 concentrations were calculated using the extinction coefficient at 280 nm of 39,225 M$^{-1}$ cm$^{-1}$. The assay was linear with time and after 10 min at 37 °C the reaction was stopped by the addition of 4 μl of 10% (v/v) acetic acid. The mixture was spotted onto a DE81 disk, washed with three successive changes of 1% acetic acid in 95% ethanol and product formation determined by scintillation counting of the dried disc[33]. If the IC$_{50}$ was determined to be in the nM range then the assay was repeated in the presence of 0–1 μM or 0–0.1 μM compound to more precisely determine the IC$_{50}$. All the experiments were repeated twice in duplicate and the data were an average ± data range. For the kinetic experiments, the assay was done either varying the pantothenate from 0–180 μM or ATP from 0–125 μM at a given concentration of PZ-2724.

The experiments mimicking the mixture of ligands present in vivo were performed under different conditions. The reaction mix for the determination of the acetyl-CoA IC$_{50}$ contained 100 mM Tris-HCl (pH 7.5), 10 mM MgCl$_2$, 1 mM ATP, 45 μM D-[1-$^{14}$C]pantothenate (specific activity 22.5 mCi/mmol), 2.5 μM PZ-2891 and 1 μg of PANK3. In the time course experiments, the reaction mixtures contained 100 mM Tris-HCl (pH 7.5), 10 mM MgCl$_2$, 1 mM ATP, 45 μM or 90 μM D-[1-$^{14}$C]pantothenate (specific activity, 22.5 mCi/mmol), 100 μM acetyl CoA ± 2.5 μM PZ-2891 and 1 μg of PANK3.

The IC$_{50}$ values for the structure–activity study were calculated by fitting the inhibition data to a one-site model of the Michaelis–Menten equation. Although this method was appropriate to measure the IC$_{50}$ with most of the pantazines, it underestimates ligand binding affinity if the concentration of protein in the assay alters the free ligand concentration. Thus, the PZ-2891 data were fit to Morrison's quadratic equation[48] (GraphPad software) that accounts for the impact of enzyme-inhibitor binding on the free concentration of inhibitor.

**Protein thermal shifts.** Protein thermal shift experiments were conducted in a mixture containing 100 mM HEPES, pH 7.0, 5X Sypro Orange dye, 10 mM MgCl$_2$, 2.5 μM PANK3 and ATP, acetyl CoA or pantazines at the indicated concentrations. Reaction mixture (100 μl) was aliquoted into optically clear 96-well PCR plates, centrifuged and placed in an ABI 7500 real time PCR system for thermal shift analysis. The temperature was ramped from 25 °C to 95 °C at a rate of 1 °C/min, and fluorescence was measured using a TAMRA filter set (Ex 560 nm and Em 582 nm). The data were plotted as fluorescence intensity as a function of temperature and were fit to the first derivative of the Boltzman sigmoidal equation using GraphPad software to determine the temperature corresponding to the denaturation of 50% of the protein. Each experiment was performed in triplicate and the melting temperatures were the average of 3 independent data sets rounded to the nearest degree. For determining the K$_{0.5}$, the change in melting temperature from all 3 data sets was plotted against concentration of the ligand, and those data were fit to the Michaelis–Menton equation using GraphPad software.

**Gel filtration chromatography.** [$^3$H]ATP (300 μM; 1.38 Ci/mmol) was incubated with PANK3 in a mixture containing 100 mM Tris pH 7.5, 10 mM MgCl$_2$, with or without 5 μM PZ-2891 on ice for 10 min, and then applied to a Superdex 75 10 300GL column (GE Lifesciences) equilibrated and eluted with 20 mM Tris-HCl, pH 7.5, 200 mM NaCl. Fractions (200 μl) were collected, and 100 μl were counted by scintillation counting. The radioactivity counts obtained were then plotted against the elution volume and compared to the absorbance values obtained at 280 nm in the elution profile. The gel filtration experiment was performed twice, and one of the experiments is represented here.

**Cell culture and radiolabeling.** HEK293T cells (ATCC, #CRL-3216) were maintained in Dulbecco's modified Eagle's medium (DMEM, Biowhittaker/Lonza) supplemented with 10% FCS. Human C3A [HepG2/C3A, derivative of HepG2] cells (ATCC #CRL-10741) were purchased from ATCC® and maintained in Eagle's minimum Essential medium (ATCC) supplemented with 2 mM glutamine, 10%

fetal bovine serum (FCS, Altanta Biologicals), 50 U/ml penicillin and 50 mg/ml streptomycin. Both cell lines were confirmed to be mycoplasma-free. HEK293T cells were used for overexpression studies and were transfected using FuGENE® 6 reagent (Promega) combined with 12 μg of the indicated plasmid cDNA construct: pcDNA3.1(−) vectors expressing either human PANK3, PANK1β, PANK2m or PANK3(E138A) according to the manufacturer's recommendations. PZ-2891, PZ-3067 or vehicle control was added 24 h post-transfection. After 24 h of treatment (total 48 h after transfection) the cells were washed with PBS and harvested and subjected to either total CoA or acetyl-CoA determinations.

For radiolabeling human C3A cells with pantothenate, medium was prepared from glucose-, pantothenate-, pyruvate-, choline chloride-, methionine- and glutamine-free DMEM (Gibco/Life Technologies/ThermoFisher) reconstituted with 1 g/l glucose, 1 mM pyruvate, 1 mg/ml choline chloride, 15 mg/ml methionine, 2 mM glutamine, and 1 mCi/ml [2,3-$^3$H]pantothenic acid (specific activity 50 Ci/mmol; American Radiolabeled Chemicals, ART 0731), 10% delipidated FBS (Cocalico Biologicals #55–0115), 50 U/ml penicillin, and 50 mg/ml streptomycin. Prior to labeling, cells were seeded at a density of 2–3 × 10$^6$ in 60 mm-dishes in reconstituted DMEM without added pantothenate and allowed to adhere overnight. The labeling was initiated by replacing the medium with 2 ml of labeling medium containing 0.1% dimethylsulfoxide (DMSO; control) or 10 μM of the compound in 0.1% DMSO. Cells were incubated for 24 h, the labeling medium was removed, and cells were washed with phosphate buffered saline and lysed by sonication in 20 mM Tris-HCl, pH 7.5, 2 mM DTT, 5 mM EDTA and 50 mM NaF. The lysates were fractionated by thin-layer chromatography after removal of the cell debris by centrifugation[11]. The samples were spotted on Silica Gel H plates and resolved with 95% ethanol:28% ammonium hydroxide (4/1, v/v). The distribution of radioactivity on the plate was quantified using a Bioscan Imager (Bioscan, Inc.) together with quantification of the total cellular radioactivity determined by a Liquid Scintillation Analyzer (PerkinElmer Tri-Carb 2910 TR). Triplicate culture dishes were labeled for each group plus a dish for each group for determination of cell counts and viability using a Nucleocounter (New Brunswick Scientific) according to the manufacturer's directions. The experiments were repeated twice in duplicate and the data represented is an overall average ± SEM.

For quantifying protein, DNA and lipid synthesis, C3A cells were first seeded at 2 × 10$^6$ in 60 mm-dishes and grown overnight in maintenance medium. Cells were washed once with phosphate buffered saline (PBS) and 2 ml of labeling medium plus radiochemical was added to start the 24 h incubation. Labeling medium consisted of 90% DMEM lacking glucose and pyruvate (Life Technologies), 10% DMEM containing 4.5 g/l glucose, lacking glutamine (Biowhittaker/Lonza) plus 10% delipidated FBS (Cocalico Biologicals). A final concentration of 5 μCi/ml $^3$H-labeled amino acid mixture (specific activity 60 Ci/mmol; American Radiolabeled Chemicals, ART 0328), 5 μCi/ml [$^3$H]thymidine (specific activity 84 Ci/mmol; American Radiolabeled Chemicals, ART 0178) or 5 μCi/ml [U-$^{14}$C]glucose (specific activity 10 mCi/mmol; American Radiolabeled Chemicals, ART 0122 A) was used to measure protein synthesis, DNA synthesis or lipid synthesis, respectively. The labeling medium contained 0.1% dimethylsulfoxide (DMSO; control) or 10 μM compound in 0.1% DMSO. Cells were harvested, washed once with PBS and harvested either by trypsinization and vacuum filtration through 0.45 micron discs (Millipore, HAWP02500) for protein and DNA determinations, or by scraping and extraction in 100 μl H$_2$O + 240 μl methanol/acetic acid 98:2 + 100 μl CHCl$_3$ for lipid determination. Filters were washed three times with PBS, air dried and counted with a Liquid Scintillation Analyzer. Radioactivity was normalized to cell numbers and viability was >95%. Cellular lipid emulsions were separated by centrifugation and the lower phases were quantified by liquid scintillation counting and normalized to cell numbers.

**Human PANK isoform distribution.** The levels of human PANK mRNA were determined by real time qPCR in C3A cells that were grown to a density of 3 × 10$^6$ cells/dish and were treated either with 10 μM PZ-2891 or DMSO for 24 h. RNA was isolated immediately using Trizol reagent according to the manufacturer's instructions (Invitrogen). The cDNA was synthesized using Super Script$^{TM}$ II, random primers and the RNA templates after the removal of genomic DNA by turbo DNAse (Ambion). Quantitative real-time PCR was performed using SYBR Green master mix in triplicate with the primers listed below. The human glyceraldehyde-3-phosphate dehydrogenase *GAPDH* (Applied Biosystems) was used as a control. All of the real-time values were compared using the $C_T$ method, where the amount of *PANK1, 2* or *3* cDNA was normalized to the housekeeping gene *GAPDH* ($\Delta C_T$) before being compared with the amount of *PANK* cDNA under DMSO condition ($\Delta\Delta C_T$), which was set as the calibrator at 1.0. The data represent an average of three experiments. The primers are listed in Supplementary Table 5.

**PANK antibody.** The *pan*-reactive anti-PANK antibody that recognized all mouse and human PANK isoforms was purified from rabbit polyclonal antiserum that was raised against pure full-length recombinant human PANK1β. Immunization of rabbits and collection of antisera were performed by Rockland Immunochemicals, Inc. according to their standard schedule. Antisera were purified by affinity chromatography on Affi-Gel cross-linked with the recombinant PanK1β protein. The purified α-PANK antibody was validated by immunoblotting serial dilutions of the purified antibody preparations against purified recombinant mouse and human

PANK1α, PANK1β, PANK2 and PANK3 proteins, and cell lysates expressing each of the isoforms (Supplementary Fig. 17). The entire blots for the cropped data in Fig. 4a and Fig. 4e are presented as Supplementary Fig. 18 and Supplementary Fig. 19, respectively.

**Cellular thermal shift**. The method was adapted from Jafari, et al[38]. HEK293T cells were transfected with 5 μg/dish pPANK3, a pcDNA3.1(−)-derived expression vector, in a 100 mm dish. After 24 h the cells were treated with the indicated concentrations of pantazine or 0.1% DMSO for 2 h at 37 °C. Four dishes were used in each assay. The cells were washed and resuspended in PBS, counted to determine viable cells and then were resuspended in PBS at $5 \times 10^5$ cells/100 μl. Aliquots (100 μl) in 0.5 ml reaction tubes were exposed to various temperatures (25, 53, 55, 57, 59, 61, 63, 65 and 67 °C) in pairs (control and compound treated) using a thermo cycler PCR machine for 3 min and then moved to room temperature for another 3 min before being flash frozen in liquid nitrogen, and stored at −80 °C. The samples were freeze-thawed 5 times, sonicated at 4 °C for 10 s, and the lysates were centrifuged at 14,000×g for 20 min to remove precipitated proteins. The soluble fractions were fractionated on a 10% bis-Tris acrylamide gel and transferred to a PVDF membrane. The blots were blocked for 1 h in 1% milk/TBS-T and then exposed to primary antibody (α-PANK) overnight at 1 μg/ml in 1% BSA/TBS-T followed by secondary antibody (anti-Rabbit AP conjugated) in 1% milk/TBS-T for 1 h at 1:5000 dilution. The blot was washed extensively and exposed to the ECF substrate for 5 min, and the bands on the dried membrane were quantified on the Typhoon FLA9500 using ImageQuant TL software (GE Healthcare). The control blot was done with samples exposed to different temperatures but were not centrifuged to remove the precipitated proteins and probed with anti-β-actin antibody (Sigma-Aldrich). All the experiments were repeated twice in duplicate and the combined data are presented (overall average ± SEM).

**Tissue and cellular CoA determinations**. Cultured cells or frozen tissues were resuspended in 2 ml cold water to which 500 μl of 0.25 M KOH was added, derivatized with monobromobimane (mBBr, Life technologies) and quantified by HPLC[23]. For acetyl-CoA determinations, C3A cells were resuspended in 1 ml water, added to 2 ml of methanol and 1 ml chloroform, and incubated on ice for 15 min. Chloroform 1.5 ml, 1.2 ml water and 30 pmol of [$^{13}C_2$]acetyl-CoA (Sigma) was added, and centrifuged at 2000×g for 10 min. The top layer was loaded on a 2-(2-pyridyl) ethyl solid phase extraction column which was equilibrated with 1 ml 50% methanol/2% acetic acid. The column was washed twice with 1 ml 50% methanol/2% acetic acid and 1 ml water. CoA and thioesters were eluted twice with 1 ml 95% ethanol containing 50 mM ammonium hydroxide. Samples were resuspended in 90% methanol containing 15 mM ammonium hydroxide.

**CoA, dephosphoCoA and phosphopantetheine determinations**. The mBBr derivatized sample was fractionated by reverse-phase HPLC using a Gemini C18 3 μm column (150 × 4.60 mm) from Phenomenex. The chromatography system was a Waters e2695 separation module with a UV/Vis and fluorescence detector and controlled by Empower 3 software. Solvent A was 50 mM potassium phosphate pH 4.6, and solvent B was 100% acetonitrile. Twenty microliters of sample was injected onto the column, and the flow rate was 0.5 ml/min. The HPLC program was the following: starting solvent mixture of 90% A/10% B, 0–2 min isocratic with 10% B, 2–6 min linear gradient from 10% B to 15% B, 6–18 min concave gradient from 15% B to 40% B, 18–23 min isocratic with 40% B, 23–25 min linear gradient from 40 to 10%, and 25–30 min isocratic with 10% B. The UV/vis detector was set at 393 nm, and the fluorescence detector was set with excitation at 393 nm and emission at 470 nm. The elution position of the mBBr-CoA, mBBr-dephospho-CoA (deP-CoA), and mBBr-phosphopantetheine (PPanSH) were determined by comparison with mBBr-CoA prepared from commercial CoA (Avanti Polar Lipids), mBBr-deP-CoA prepared from commercial deP-CoA (Sigma-Aldrich), and mBBr-PPanSH prepared from a Nudt7-mediated hydrolysis of mBBr-CoA. The areas under the mBBr-derivatized CoA, deP-CoA and PPanSH peaks were integrated and compared to known concentrations of the mBBr-CoA standard.

**Acetyl-CoA measurement by mass spectrometry**. Mass spectrometry of acetyl-CoA was performed using a Finnigan TSQ Quantum (Thermo Electron) triple-quadrupole mass spectrometer. The instrument was operated in positive mode using single ion monitoring (SIM) neutral loss scanning corresponding to the loss of the phosphoadenosine diphosphate from CoA species. The ion source parameters were as follows: spray voltage 4000 V, capillary temperature 250 °C, capillary offset −35 V, sheath gas pressure 10, auxiliary gas pressure 5, and tube lens offset was set by infusion of the polytyrosine tuning and calibration in electrospray mode. Acquisition parameters were as follows: scan time 0.5 s, collision energy 30 V, peak width Q1 and Q3 0.7 FWHM, Q2 CID gas 0.5 mTorr, source CID 10 V, neutral loss 507.0 m/z, SIM mass of 810 m/z with a scan width of 8 m/z to capture the signal from cellular acetyl-CoA and the [$^{13}C$]acetyl-CoA (Sigma) internal standard.

**PZ-2891 extraction and quantification by LC/MS/MS**. Plasma (20 μl) was added to 100 μl acetonitrile containing 0.6 μM warfarin to a final concentration of 0.5 μM. The samples were incubated on ice for 30 min. Samples were spun at 3500×g for 10

min to pellet debris, and the supernatant was transferred to a glass vial. A PZ-2891 standard curve was created by spiking in known concentrations of PZ-2891 into 20 μl of plasma from a control mouse and following the above procedure. Tissue was homogenized in 2 ml of 80% methanol containing 0.1 μM warfarin, and incubated at −80 °C for 4 h. Samples were spun at 3500×g for 10 min to pellet debris, supernatant was transferred to a glass tube and dried down using a Savant SPD1010 Speed-Vac (Thermo Scientific) overnight. Samples were resuspended in 400 μl of 80% acetonitrile to a final concentration of 0.5 μM and transferred to a glass vial.

PZ-2891 was analyzed using a Shimadzu Prominence UFLC attached to a Sciex QTrap 4500 equipped with a Turbo V ion source. Samples (5 μl) were injected onto an XSelect® HSS C18, 2.5 μm, 3.0 × 150 mm column (Waters) using a flow rate of 0.25 ml/min. Solvent A was 0.1% formic acid in water, and Solvent B was acetonitrile + 0.1% formic acid. The HPLC program was the following: starting solvent mixture of 50% B, 0–0.5 min isocratic with 50% B; 0.5–1.5 min linear gradient to 95% B; 1.5–20 min isocratic with 95% B; 20–21 min linear gradient to 50% B; 21–25 min isocratic with 50% B. The QTrap 4500 was operated in the positive mode, and the ion source parameters were: ion spray voltage, 5500 V; curtain gas, 30 psi; temperature, 450 °C; collision gas, medium; ion source gas 1, 30 psi; and ion source gas 2, 40 psi. The MRM transition for PZ-2891 was 350.2 / 190.0 m/z and warfarin was 309.1 / 163.0 m/z both with a declustering potential, 65 V and collision energy, 30 V. The system was controlled by the Analyst® software (Sciex) and analyzed with MultiQuant™ 3.0.2 software (Sciex).

**Pantothenate extraction and quantification by HPLC/MS/MS**. Plasma (10 μl) was added to 800 μl methanol and 190 μl water along with 200 pmol (β-alanyl-$^{13}C_3$$^{15}$N)-pantothenate ($^{13}C^{15}$N-Pan, Sigma). Tissue (30–40 mg) was homogenized in 2 ml of 80% methanol, and 500 pmol of $^{13}C^{15}$N-Pan was added. The samples were incubated at −80 °C for 4 h. Samples were spun at 3500×g for 10 min to pellet debris, and supernatant was transferred to a new tube and dried using a SpeedVac overnight. Samples were resuspended in water + 0.1% formic acid and spun through a Spin-X Centrifuge Tube Filter (0.22 μm Cellulose Acetate, Costar).

Pantothenate was analyzed using a Shimadzu Prominence UFLC attached to a QTrap 4500 equipped with a Turbo V ion source (Sciex). Samples were injected onto an XSelect® HSS C18, 2.5 μm, 2.1 × 150 mm column using a flow rate of 0.2 ml/min. Solvent A was 0.1% formic acid in water, and Solvent B was acetonitrile + 0.1% formic acid. The HPLC program was the following: starting solvent mixture of 1% B, 0–6 min linear gradient to 60% B; 6–13 min linear gradient to 100% B; 13–16 min isocratic with 100% B; 16–17.5 min linear gradient to 1% B; 17.5–20 min isocratic with 1% B. The QTrap 4500 was operated in the positive mode, and the ion source parameters were: ion spray voltage, 5500 V; curtain gas, 30 psi; temperature, 400 °C; collision gas, medium; ion source gas 1, 20 psi; and ion source gas 2, 25 psi. The MRM transition for pantothenate was 220.0 / 184.1 m/z and [$^{13}C^{15}$N]pantothenate was 224.0 / 188.1 m/z both with a declustering potential, 35 V and collision energy, 22 V. The system was controlled by the Analyst® software (Sciex) and analyzed with MultiQuant™ 3.0.2 software (Sciex). Intracellular pantothenate was measured as pmoles/mg wet weight.

**Animal ethics statement**. All procedures were reviewed and approved by the St. Jude Children's Research Hospital Institutional Animal Care and Use Committee.

**Mouse efficacy testing**. C57Bl6/J mice (8-week-old) were purchased from Jackson Laboratory and were fed either a regular diet (autoclavable rodent breeder diet, Labdiet) or an isocaloric, pantothenic acid-supplemented diet (1000 ppm) for 2 weeks prior to the experiment. The mice were maintained at room temperature 72° ± 2 °F, humidity 50 ± 10% and a 14 h light /10 h dark cycle with the dark cycle starting at 18:00 h. Water was supplied ad libitum. The mice were randomized into the treatment arms to achieve a normal weight distribution. PZ-2891 was formulated in 30% Captisol[49], and was administered by oral gavage at 12 h intervals for 5 doses. The mice were euthanized and tissues harvested 4 h after the last dose. The tissue samples were used for total CoA, pantothenate and pantazine determinations. Blood was collected from euthanized animals, plasma or serum was prepared and stored frozen until analysis. Organs, including liver, forebrain, and hindbrain were quickly excised from euthanized animals and immediately flash frozen in liquid $N_2$ or immersed in RNALater® (Qiagen) overnight prior to freezing. Forebrain and hindbrain regions were identified[50]. Total CoA was determined using 20–50 mg of tissue (liver, forebrain or hindbrain) homogenized in 2 ml of 1 mM KOH. The pH was adjusted to 12.0 with 0.25 M KOH, and incubated at 55 °C for 2 h[23]. The pH of the sample was adjusted to 8.0, and the samples were derivatized with monobromobimane (mBBr) and analyzed by HPLC equipped with a fluorescence detector.

**Generation of SynCre Pank1,Pank2 neuronal knockout mice**. Generation of the Pank1$^{fl/fl}$ and the Pank2$^{fl/fl}$ mice was reported previously[12,13]. The SynCre transgene originated in B6.Cg-Tg(Syn1-cre)671Jxm/J transgenic mice (The Jackson Laboratory) that express the Cre recombinase driven by the synapsin1 (Syn) promoter. Pank1$^{fl/fl}$,Pank2$^{fl/+}$ SynCre+ (or Pank1$^{fl/+}$,Pank2$^{fl/fl}$ SynCre+) females were mated with Pank1$^{fl/fl}$,Pank2$^{fl/fl}$ SynCre− males to obtain PANK1$^{fl/fl}$, PANK2$^{fl/fl}$ SynCre+ progeny that have both Pank1 and Pank2 conditionally

deleted in neuronal tissues. Control littermate mice had the *Pank1*[fl/fl],*Pank2*[fl/fl] *SynCre−* genotype. PCR genotyping primer pairs and products are listed in Supplementary Table 4. PCR analysis was used to genotype tail biopsies where a 338 bp product indicated the floxed *Pank1* allele, a 332 bp product indicated the floxed *Pank2* allele, and a 285 bp product indicated the presence of the *Cre* transgene. RNA was isolated from cryo-preserved liver or brain tissue. Synthesis of first-strand cDNA was obtained by reverse transcription using SuperScript[TM] RNase H reverse transcriptase, the RNA templates and random primers. Quantitated real-time PCR was performed in triplicate using the ABI Prism®7700 Sequence Detection System with the primers listed in Supplementary Table 5. The Taqman human GAPDH (Applied Biosystems) were used as controls. All of the values were compared using the $C_T$ method[51], and the amount of cDNA ($2^{-\Delta CT}$) was calculated relative to human glyceraldehyde-3-phosphate dehydrogenase mRNA. Neuron-specific deletion of the *Pank1*- and *Pank2*-floxed alleles was confirmed in brains by the presence of a 218 bp and 176 bp product, respectively, that were absent in liver (Supplementary Figure 14). Incomplete deletion of the floxed alleles in brain was indicated by coincidence of the *Pank1*-floxed and *Pank1*-deleted PCR products, and the *Pank2*-floxed and *Pank2*-deleted PCR products. Cell types other than neurons contributed to the undeleted, residual *Pank1*- and *Pank2*-floxed genes in brain. Wild-type matched control animals were derived from breeding pairs that were heterozygous for both *Pank1*-floxed and *Pank2*-floxed alleles and lacked the *SynCre* transgene. Following genotyping, *SynCre+* and *SynCre− Pank1*-floxed and *Pank2*-floxed mice regardless of sex were randomly enrolled into the treatment or control arms of the PZ-2891 trial as they emerged from the breeding program.

**Open field locomotion**. Individual mice were placed in an open rectangular arena (36.8 cm × 43.2 cm) for 5 min during the light phase and motor activity was evaluated using a video tracking system provided by HVS Image with associated 2100 Plus software (San Diego, CA, USA). Each subject was placed in the center of the arena under standard overhead lighting and the total distance traveled and the percentage of time in motion were recorded. The spontaneous activity of each subject in a novel open field provided a general measure of motor function of the *SynCre* (+/−) *Pank1, Pank2* floxed mice, age 45 days, treated or not with PZ-2891. The behaviors of age-matched control animals, treated with PZ-2891 and untreated, were quantified and compared.

**Crystallization and structure determination**. PANK3 with two amino acids (DD) added to the carboxy-terminus was expressed, purified and crystalized[9]. The crystals of the PANK3•AMPPNP•Mg$^{2+}$•pantothenate complex[9] were soaked in mother liquor (0.2 M ammonium acetate, 0.1 M citrate, pH 5.6, 50 mM MgCl₂, 32% polyethylene glycol (4 K), 4% DMSO, 10 mM AMPPNP (adenosine 5′-(β,γ-imido)triphosphate) and 1 mM PZ-2891 for two days. The crystals were cryoprotected with 29% ethylene glycol. Diffraction data were collected at the SER-CAT beam line 22-ID at the Advanced Photon Source, and processed using HKL2000[52]. The structure was solved by molecular replacement using the PANK3 structure (PDB ID: 3SMS) and the program PHASER[53]. The structure was refined and optimized using PHENIX[54] and COOT[55], respectively. The refined structure was validated using MolProbity[56]. The atomic coordinates and structure factors have been deposited in the Protein Data Bank as PDB entry 6B3V. The data collection and refinement statistics are presented in Supplementary Table 3. Ramachandran statistics (favored/allowed/outliers (%)) 98.28/1.72/0.0 for the PZ-2891 complex. All structures were rendered with PyMOL (version 1.8, Schrödinger, LLC). For co-crystallization of the PANK3•AMPPNP•Mg$^{2+}$•PZ-2891 complex, the protein solution was incubated with 25 mM AMPPNP (adenosine 5′-(β,γ-imido)triphosphate), 25 mM MgCl₂, and 1.25 mM PZ-2891 (final 5% DMSO) to form a complex. The ternary complex was crystallized using the hanging-drop vapor-diffusion method at 18 °C. Crystals were grown by mixing 1.5 μl of complex solution with 1.0 μl of reservoir solution containing 0.1 M HEPES, pH 7.5, 10% isopropanol, 20% PEG 4 K, and 0.1 M guanidine hydrochloride.

**Statistics**. All statistical calculations were performed using GraphPad Prism. Tests between two groups used two-tailed unpaired Student's *t* test. Data are presented as means ± SD, and the *p* values and number of biological replicates are provided in the figures. The log-rank (Cox-Mantel) test was used to analyze the survival data. The pre-established exclusion criteria in the Grubbs test was used to determine if there were outliers. Sample size (at least 5 mice per group) was chosen based on pilot experiments measuring the normal distribution of CoA levels in mouse tissues (between 4% and 10%). No mice were excluded from the study, and there was no blinding.

**General chemical methods**. Unless otherwise noted, all reactions were carried out in flame-dried glassware under a static nitrogen atmosphere with anhydrous solvent. All reagents were obtained from commercially available sources and used without purification. Purification was handled by flash Silica Gel chromatography using a Biotage SP-1 chromatography system and SNAP KP-Sil cartridges. All final compounds were purified to >95% purity indicated as averages of the total

wave count (TWC) and the ELSD readings in LC/MS chromatogram (column: Acquity BEH C18). [1]H and [13]C spectra were recorded using 400 or 500 MHz, using CDCl₃ as solvent. The chemical shifts are reported in parts per million (ppm) relative to CDCl₃.

**Synthesis of PZ-2891 and PZ-3067**. 1-Boc-piperazine (200 mg, 1.43 mmol, 1 eq) and 6-chloropyridazine-3-carbonitrile (320 mg, 1.72 mmol, 1.2 eq) were dissolved in anhydrous acetonitrile (6 ml) in microwave vial. Triethylamine (1 ml, 7.17 mmol, 5 eq) was added to it dropwise. The vial was capped and the reaction mixture was heated at 160 °C for 30 min using microwave irradiation. The mixture was cooled to room temperature and concentrated to dryness. The crude product was dissolved in excess dichloromethane and washed with 1 M HCl followed by water and brine. The organic phase was dried over MgSO₄, and concentrated. The purification was performed by flash column chromatography using hexane/ethyl acetate gradient to obtain tert-butyl 4-(6-cyanopyridazin-3-yl)piperazine-1-carboxylate (S1, 378 mg, 91% yield); [1]H NMR (400 MHz, Chloroform-*d*) δ 7.47 (d, *J* = 9.6 Hz, 1 H), 6.84 (d, *J* = 9.6 Hz, 1 H), 3.80 (m, 4 H), 3.64–3.55 (m, 4 H), 1.49 (s, 9 H); [13]C NMR (101 MHz, CDCl₃) δ 158.65, 154.69, 130.85, 129.75, 116.81, 109.99, 80.73, 44.39, 31.09, 28.53; LCMS (m/z): 290.16 (M$^+$ + H, observed), 289.15 (M$^+$, calculated).

To deprotect the Boc functionality, S1 (105 mg, 0.363 mmol) was treated with trifluoroacetic acid and dichloromethane (1:1, 3 ml) at room temperature. After 1 h, reaction was monitored using UPLC and the mixture was concentrated to dryness. The crude (6-(piperazin-1-yl)pyridazine-3-carbonitrile) (S2) was used directly for the next step. 6-(piperazin-1-yl) pyridazine-3-carbonitrile (S2 crude, 0.363 mmol) was dissolved in 10 ml anhydrous dichloromethane. HATU (207 mg, 0.545 mmol, 1.5 eq) and DIPEA (0.13 ml, 0.726 mmol, 2 eq) were added to it. The reaction mixture was stirred overnight at room temperature. The reaction mixture was then diluted with 1 M HCl solution and extracted with methylene chloride. The organic phase was washed with water, brine, dried over MgSO₄, filtered and concentrated. The crude residue was purified by flash column chromatography on a Biotage SP-1 chromatography system using hexane/ethyl acetate to obtain 6-(4-(2-(4-isopropylphenyl)acetyl)piperazin-1-yl)pyridazine-3-carbonitrile (PZ-2891, 98 mg, 77% yield for 2 steps); [1]H NMR (400 MHz, Chloroform-*d*) δ 7.45 (d, *J* = 9.6 Hz, 1 H), 7.19 (m, 4 H), 6.81 (d, *J* = 9.6 Hz, 1 H), 3.82 (m, 2 H), 3.76 (s, 2 H), 3.75–3.66 (m, 4 H), 3.63 (m, 2 H), 2.88 (m, 1 H), 1.23 (d, *J* = 6.9 Hz, 6 H); [13]C NMR (126 MHz, CDCl₃) δ 170.27, 158.54, 147.93, 131.79, 130.91, 130.02, 128.59, 127.16, 116.68, 110.05, 45.42, 44.54, 44.01, 41.12, 40.85, 33.87, 24.11; HRMS (m/z): 350.1983 (M$^+$ + H, observed), 349.1903 (M$^+$, calculated). [1]H-NMR and [13]C-NMR spectra of PZ-2891 are in Supplementary Fig. 20. Liquid chromatography analysis and mass spectrum of PZ-2891 are in Supplementary Fig. 21.

6-(Piperazin-1-yl)pyridazine-3-carbonitrile (S2 crude, 0.287 mmol) was dissolved in anhydrous dichloromethane (10 ml) which was followed by addition of 2-bromo-1-(4-isopropylphenyl)ethan-1-one (104 mg, 0.430 mmol, 1.5 eq) and DIPEA (0.25 ml, 1.434 mmol, 5 eq). The reaction mixture was stirred at room temperature for 3 h. After the reaction was completed, 1 M HCl was added and it was extracted with dichloromethane. The organic phase was dried with MgSO₄, and the solvent was removed. The purification was performed by flash column chromatography using hexane/ethyl acetate gradient to obtain product 6-(4-(2-(4-isopropylphenyl)-2-oxoethyl)piperazin-1-yl)pyridazine-3-carbonitrile (PZ-3067, 92 mg, 92% yield). [1]H NMR (400 MHz, Chloroform-*d*) δ 7.93 (d, *J* = 8.3 Hz, 2 H), 7.44 (d, *J* = 9.6 Hz, 1 H), 7.33 (d, *J* = 8.3 Hz, 2 H), 6.83 (d, *J* = 9.5 Hz, 1 H), 3.89 (m, 6 H), 2.98 (h, *J* = 6.9 Hz, 1 H), 2.77 (m, 4 H), 1.27 (d, *J* = 6.9 Hz, 6 H); [13]C NMR (101 MHz, CDCl₃) δ 195.49, 158.63, 155.37, 133.78, 130.73, 129.42, 128.46, 126.95, 116.97, 109.83, 63.94, 52.94, 44.55, 34.47, 23.79; HRMS (m/z): 350.1985 (M$^+$ + H, observed), 349.1903 (M$^+$, calculated).

**Synthesis of PZ-2789**. 6-(Piperazin-1-yl)nicotinonitrile (102 mg, 0.542 mmol, 1 eq) was dissolved in dichloromethane (5 ml) and 1-(tert-butyl)-4-iso-cyanatobenzene (104 mg, 0.596 mmol, 1.1 eq) was added to it which was followed by dropwise addition of triethylamine (0.08 ml, 0.596 mmol, 1.1 eq). The reaction mixture was stirred at room temperature and the reaction was monitored using LC-MS. After the reaction was completed (within 1 h), the reaction mixture was evaporated to dryness. The crude residue was purified by flash column chromatography using hexane/ethyl acetate gradient to obtain N-(4-(tert-butyl)phenyl)-4-(5-cyanopyridin-2-yl)piperazine-1-carboxamide (PZ-2789, 165 mg, 84% yield); [1]H NMR (400 MHz, Chloroform-*d*) δ 8.46 (dd, *J* = 2.4, 0.8 Hz, 1 H), 7.69 (dd, *J* = 9.0, 2.3 Hz, 1 H), 7.38–7.32 (m, 2 H), 7.30 (d, *J* = 2.1 Hz, 2 H), 6.63 (dd, *J* = 9.0, 0.9 Hz, 1 H), 6.31 (s, 1 H), 3.83 (m, 4 H), 3.74–3.65 (m, 4 H), 1.32 (s, 9 H); [13]C NMR (101 MHz, CDCl₃) δ 155.21, 152.77, 146.72, 140.23, 135.95, 125.99, 120.16, 105.84, 97.27, 43.93, 43.28, 34.45, 31.54; HRMS (m/z): 364.2152 (M$^+$ + H, observed), 363.2059 (M$^+$, calculated).

## Data availability

The PANK3•AMPPNP•Mg$^{2+}$•PZ-2891 crystal structure and diffraction data have been deposited with the worldwide protein data bank under accession code 6B3V [https://doi.

org/10.2210/pdb6B3V/pdb]. The data that support the findings of this study are available from the corresponding authors on request.

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

## Acknowledgements

This work was supported by a sponsored research grant from CoA Therapeutics, Cancer Center core grant CA21765 and the American Syrian Lebanese Associated Charities. We thank the St. Jude Protein Production Facility for protein expression/purification, the St. Jude Hartwell Center for DNA sequencing, Louis Richmond for animal experiments, Caroline Pate for CETSA experiments, Katie Creed for biochemical and biophysical assays, Karen Miller for animal experiments, protein purification and molecular biology, Jina Wang for cell culture experiments, Brett Waddell of the Molecular Interaction Shared Resource for surface plasmon resonance analysis, and Lei Yang of the High Throughput Analytical Core for ADME analysis.

## Author contributions

All authors contributed to formulating the conclusions, writing the manuscript and approved the text, tables and figures. L.K.S. and R.E.L. designed and interpreted the chemical biology experiments. L.K.S. synthesized the compounds and prepared the formulation for animal studies. S.W.W. and M-K.Y. designed and interpreted the structural biology experiments. M.W.F. performed the CoA measurements and mass spectrometry. C.O.R. and C.S. designed and interpreted the biochemistry experiments, S. J. derived the animal model and S.J. and C.S. designed and interpreted the cell and animal experiments.

## Additional information

**Competing interests:** The authors declare the following competing interests. L.K.S., C. S., M.-K.Y., C.O.R., R.E.L. and S.J. are inventors on a pending patent application (PCT/US17/39037) "Small molecule modulators of pantothenate kinases" held by St. Jude Children's Research Hospital that covers the pantazine chemical series disclosed in this manuscript. S.J., R.E.L., C.O.R. and S.W.W. received research funding from CoA Ttherapeutics. The authors declare no other competing interests.

