## [Peer Review File · Nature Communications]

Reviewers' Comments:

Reviewer #1:

Remarks to the Author:

This study characterized an activator of pantothenate kinase (PANK) and examined its ability to increase CoA levels in the brain of a mouse model of mice with a PANK2 gene deficiency. The PANK activator developed, PZ-2891, was able to increase life span of mice with a CoA deficiency.

Critique:

The majority of this paper describes the development and characterization of PZ-2891, which activates PANK3 at lower concentrations and inhibits it at higher concentrations. The authors provide convincing data as to how PZ-2891 acts on PANK3, the pharmacokinetic properties of PZ-2891, and data on the bioavailability of PZ-2891 in mice. I do not have any major concerns with this aspect of the study, although it is not clear to me if in Nature Communications is the most appropriate journal for the publication of such an article.

The potential for using this compound to treat PANK2 deficiency disorders is of interest, although I have concerns with the appropriateness of the mouse model as a model of the clinically observed mutations in PANK2, and the limited efficacy of PZ-2891 in reversing the negative phenotype seen in the mouse model. Curiously mice with a PANK2 deletion do not develop the neurodegeneration seen in humans with a PANK2 mutation. As a result, the authors use a PANK1 -/- which also develop a PANK2 deficiency. These mice die within 20 days of birth, and it is not clear how representative they are of the human mutations in PANK2 that develop neurodegeneration. Regardless, the use of PZ-2891 only extended the lifespan of these mice by 10 days. As a result, the concept that PZ-2891 is a novel therapeutic approach to treat PANK2 deficiency is not convincing.

Specific Comments:

- 1) The characterization of PZ-2891 as a PANK activator was primarily characterized on PANK3. Very little information is provided as to what effect this compound has on PANK1 or PANK2.
- 2) The mouse model of PANK1-/- did result in a decrease in forebrain PANK2. It is not clear, however, what effect this deletion has on PANK3 expression. It is also not clear what contribution PANK3 had in maintaining forebrain CoA levels in the PANK1-/- mice.
- 3) While treatment of PANK1-/- mice resulted in an increase in lifespan by 10 days, curiously no data is provided as to whether this was accompanied by an increase in forebrain CoA levels. Instead they show that PZ-2891 increases forebrain in PANK2-/- mice, which do not possess the neurodegeneration phenotype. I believe data on forebrain CoA levels in PANK1-/- mice is critical to support the concept that the beneficial effects of PZ-2891 are due to increasing brain CoA levels.
- 4) The CoA levels in the brain and liver seem to vary greatly between mouse models. For instance, in Figure 7 the liver CoA levels seem much higher than the liver CoA levels in Figure 6, while the forebrain CoA levels in Figure 7 seem lower than the forebrain CoA in Figure 6. The question arises as to what constitutes a CoA deficiency in these mouse models?
- 5) Figure S12: It is proposed that PZ-2871 can restore the CoA deficiency in PANK2-/- mice. However, comparison of CoA levels to WT mice (Figure 6) does not really suggest that a CoA deficiency exists in the PANK2-/- mice.
- 6) Figure 2-5: It is not clear what the sample sizes are in these figures.
- 7) Figure 5: It is not clear what effect pantothenate supplementation has on brain CoA levels in these mice.
- 8) Figure 6I: The increase in hindbrain CoA was less dramatic with higher doses of PZ-2891. Was this due to inhibition of PANK at higher concentrations? Why did this not occur in other tissues such as the liver?

Reviewer #2:

Remarks to the Author:

Here, this referee reports only on the crystallographic part of the manuscript by Lalit Kumar Sharma *et al*.

The authors describe the crystal structure of a complex of one of the pantothenate kinases (PANK3) from human with AMPPNP and a PANK activator PZ-2891, (PANK3•AMPPNP•Mg²⁺•PZ-2891). In their crystal structure, PZ-2891 binds across the PANK3 dimer interface, in which PZ-2891 occupies the acetyl CoA and the pantothenate binding pockets simultaneously to make a stable PANK3•AMPPNP•Mg²⁺•PZ-2891 complex.

According to the Full wwPDB X-ray Structure Validation Report on the structure (PDB ID : 5TL2), the quality of the structure is sufficiently high. Data and refinement statistics are good. The *F_o-F_c* simulated annealing omit map (green mesh in fig. S3) looks good. The interactions of PZ-2891 with PANK3 are well presented in Fig. 2.

They soaked the crystals of the PANK3•AMPPNP•Mg²⁺•pantothenate complex with 1 mM PZ-2891 for two days. I wonder if cocrystallization rather than soaking makes any difference in the structure. Do the authors have any opinion on this point?

Reviewer #3:

Remarks to the Author:

The paper by Sharma *et al* identified a molecule denominated PZ-2891, which acts as a potent allosteric PANK activator that has oral bioavailability and crosses the blood brain barrier. The study demonstrated that PZ-2891 oral administration to mice increased CoA levels in both the liver and brain, and extended the lifespan in a mouse model of brain CoA deficiency. The Authors claimed this could be a possible therapeutic approach for neurodegenerative diseases caused by mutations of PANK2 gene responsible for PKAN.

The paper is interestingly but there are several weaknesses.

-In the abstract the Authors stated: "PZ-2891, a potent allosteric PANK activator that has oral bioavailability and crosses the blood brain barrier". Which are the experimental evidences supporting these to relevant aspects? These two points must be further discussed in the manuscript by also citing available literature or original data produced by the Authors.

-In the Introduction the sentence "The PANK2 gene is abundant in human neuronal tissues and the majority of the mutations associated with PKAN result in the expression of truncated or mutant PANK2 proteins with little or no catalytic activity" should be modified since there are several mutations associated with increased PANK2 activity. Moreover, Pank2 was also reported in mouse mitochondria by other authors. These papers must be reported and cited in the bibliography.

-In the results, chapter "PZ-2891 increases intracellular CoA levels in cultured cells", there is a certain grade of confusion about transfected and un-transfected cells, C3A and 293T cells and it is difficult to follow the data. Why different cell types were used? Which is the rationale of expressing a mature form of PANK2 in the experiment reported in Fig 4E? This protein can't reach its "physiological" localization without the mitochondrial targeting signal (MTS). So this experiment should be performed with the PANK2 cDNA containing the MTS to see if the results are different.

-The last few sentences of this chapter do not take into account several experimental evidences already published:

"CoA itself was reported to raise intracellular CoA levels": Srinivasan *et al* (Nat Chem Biol, 2015) demonstrated that CoA is converted into 4PP and is able to rescue several phenotypes in cells and *Drosophila* genetic models of PKAN. Specifically, CoA is able to rescue a null fumble mutant as well as a hypomorphic mutant, thus indicating it is converted into 4PP and then back to CoA since in *Drosophila* there is only one PANK and it is not possible for other PANKs to compensate.

"Pantetheine was not stable in serum (Fig. S11B)": this was already demonstrated by Srinivasan et al and this ref should be added.

"Although CoA was stable in serum, in the presence of cells it rapidly broke down to pantothenate (Fig. S11C). Orally-administered CoA was not detected in the plasma, but it significantly elevated plasma pantothenate levels (Fig. 11D)": these data must be supported by biochemical evidence that CoA is directly converted into pantothenate. The Authors should at least measure 4PP to understand if this is an intermediate in this conversion. According to what is known about the biosynthetic pathway of CoA this observation should imply that the pathway can also go in the backward reaction. A biochemical demonstration supporting this conversion must be provided, since this could be a major and novel achievement.

- "Thus, the conversion of CoA to pantothenate in cells and animals accounts for the reversal of hopantenate toxicity": as already mentioned there are several additional models demonstrating that CoA can reverse pathological phenotype in genetic models and not only in model of hopantenate toxicity.

Specifically:

1) Srinivasan et al demonstrated that a Drosophila genetic model is rescued by CoA addition and this could be obtained only through CoA conversion into 4PP and not into pantothenate since the only PANK enzyme is absent.

2) Orellana et al (EMBO Mol Med 2016) demonstrated that CoA was able to rescue pathological phenotypes obtained in neurons directly derived from PKAN patients, a faithful model of disease directly generated from patients' fibroblasts in which expression of PANK2 was abolished by the presence of specific mutations. Furthermore, both PANK2 re-expression in PKAN neurons and CoA addition to the culture media were able to rescue pathological phenotypes. It is true that pantothenate was not used in these experiments to evaluate its possible rescue capacity, but it is also true that pantothenate is always present in the culture media of neurons. So, a possible scenario would be either that pantothenate should be added in "huge amount" or, if present at adequate concentration, it would have prevented the expression of pathological phenotypes, which are specifically connected to the absence of PANK2.

3) There are also two models of Zebrafish, one for Pank2 (Zizioli et al, 2016) the other for Coasy (Khatri et al, 2016), suggesting that CoA addition is able to rescue pathological phenotypes.

-In figure 5 (B) liver CoA levels in mice maintained on normal diet and treated with 30 mg/kg of PZ-2891 without or with a 200 mg/kg pantothenate supplement were presented.

The Authors should also measure levels of CoA in the brains of the same animals. Moreover, the legend of this figure contains several mistakes.

It should also be mentioned that plasma pantothenate level was elevated in one PKAN patient taking oral calcium pantothenate as reported by Leoni et al 2012. How could this be reconciled with the observation obtained in wild-type animals in Fig. 5C?

-In Fig S9 again no measurement of CoA levels in the brains is reported. Moreover, it is not clear why phosphopantetheine level is reduced in mice livers but not in the cells. Phosphopantetheine is misspelled in panel C.

-In figure S11 it is compulsory to also measure 4-phosphopantetheine and/or to biochemically demonstrate that CoA is directly converted into pantothenate. I would suggest using a labelled CoA and evaluating by Mass Spec the different intermediates in both the cells and mice.

-The most interesting part of this manuscript concerned the usage of the double Pank1^{-/-} Syn-Pank2^{-/-} mouse model as a model of CoA deficiency. However, these experiments lack CoA measurement in the liver and brain of PZ-2891 treated animals. The authors should provide these measurements. Moreover, they should explain why animals die after 26 days of treatment, even though two animals survive longer. Again, in this last case it is not clear if after 66 or 119 days of treatment mice were euthanized or spontaneously died.

It is important to discuss this point in light of what was previously reported by Corbin et al 2017, that excess of CoA level reduces skeletal muscle performance and strength. Do these mice die because of respiratory distress due to diaphragm insufficiency?

Point-by-Point Response

Reviewers' comments:

Reviewer #1 (Remarks to the Author):

This study characterized an activator of pantothenate kinase (PANK) and examined its ability to increase CoA levels in the brain of a mouse model of mice with a PANK2 gene deficiency. The PANK activator developed, PZ-2891, was able to increase life span of mice with a CoA deficiency.

Critique:

The majority of this paper describes the development and characterization of PZ-2891, which activates PANK3 at lower concentrations and inhibits it at higher concentrations. The authors provide convincing data as to how PZ-2891 acts on PANK3, the pharmacokinetic properties of PZ-2891, and data on the bioavailability of PZ-2891 in mice. I do not have any major concerns with this aspect of the study, although it is not clear to me if in Nature Communications is the most appropriate journal for the publication of such an article.

The potential for of using this compound to treat PANK2 deficiency disorders is of interest, although I have concerns with the appropriateness of the mouse model as a model of the clinically observed mutations in PANK2, and the limited efficacy of PZ-2891 in reversing the negative phenotype seen in the mouse model. Curiously mice with a PANK2 deletion do not develop the neurodegeneration seen in humans with a PANK2 mutation. As a result, the authors use a PANK1 -/- which also develop a PANK2 deficiency. These mice die within 20 days of birth, and it is not clear how representative they are of the human mutations in PANK2 that develop neurodegeneration. Regardless, the use of PZ-2891 only extended the lifespan of these mice by 10 days. As a result, the concept that PZ-2891 is a novel therapeutic approach to treat PANK2 deficiency is not convincing.

- Generating mouse models that accurately reflect human neurodegeneration is challenging. PKAN is thought to arise from a brain CoA deficiency due to mutations that inactivate or attenuate PANK2 activity. Knocking out the mouse *Pank2* gene does not recapitulate the phenotypes of PKAN, perhaps because *Pank1* is expressed at a relatively higher level in murine brain compared to human. However, mouse models are essential for the preclinical evaluation of therapeutics, and we have generated over recent years a series of mouse models of brain CoA deficiency that range in severity. Although these models may not exhibit or mimic all the human disease characteristics, any PKAN therapeutic must cross the blood-brain barrier to elevate CoA. These models represent the only available preclinical tool to test the efficacy of potential drugs.
- Although we think that the *Pank1*(global)/*Pank2*(neuron-selective) knockout mouse model presented in the first submitted manuscript illustrates the biological activity of PZ-2891, the reviewers were concerned about the severity of the phenotype. We have now completed experiments with a next-generation model that directly addresses this concern. The new mouse model has both *Pank1* and *Pank2* as 'floxed' alleles, which in the presence of the *SynCre* transgene, are deleted only in neurons (new Supplementary Figs. 15 and 16). A new Fig. 7 describes the phenotype and treatment of these animals. The knockout animals fail to gain weight, have a brain CoA deficiency, and survive an average of 52 days (Fig. 7a,b,c,d). PZ-2891 therapy elevates brain CoA (Fig. 7c,d), the animals gain weight (Fig. 7a), and there is a very significant increase in their life span, from 52 to 150 days, with 5 of 11 mice still alive when the experiment was terminated at six months (Fig. 7b). Importantly, the neuron-selective *Pank1*/*Pank2* knockout mice have a severe defect in locomotion that is ameliorated by PZ-2891 therapy (Fig. 7e,f; new Supplementary Fig. 17). In our opinion,

these data provide very compelling evidence for the potential of pantazines as PKAN therapeutics.

Specific Comments:

1) *The characterization of PZ-2891 as a PANK activator was primarily characterized on PANK3. Very little information is provided as to what effect this compound has on PANK1 or PANK2.*

- We show in the HEK293T cell expression studies (Fig. 4e) that all three isoforms are activated by PZ-2891. In the revised manuscript, we added a new figure (new Supplementary Fig. 1) that shows the inhibition of all three purified mouse and human pantothenate kinase isoforms by PZ-2891. PanK1 is less sensitive to PZ-2891 than PanK2 or PanK3. These data are described at the beginning of the second Results section.

2) *The mouse model of PANK1^{-/-} did result in a decrease in forebrain PANK2. It is not clear, however, what effect this deletion has on PANK3 expression. It is also not clear what contribution PANK3 had in maintaining forebrain CoA levels in the PANK1^{-/-} mice.*

- The Cre-mediated gene deletion occurs only with 'floxed' genes. In the former mouse model, *Pank2* was 'floxed' but *Pank3* was not 'floxed.' In the new mouse model, both the *Pank1* and *Pank2* genes are 'floxed' and both genes are deleted in neurons.
- We did not observe expression changes of *Pank3* in response to PZ-2891 (Supplementary Fig. S9). We have not found *Pank3* genetic expression compensation to arise from the deletion of *Pank1* or *Pank2* in our previously published mouse models. PanK3 would be the only intact isoform expressed in neuronal cells with deleted *Pank1* and *Pank2* genes. The new Supplementary Fig. 16 provides data that illustrate this point.

3) *While treatment of PANK1^{-/-} mice resulted in an increase in lifespan by 10 days, curiously no data is provided as to whether this was accompanied by an increase in forebrain CoA levels. Instead they show that PZ-2891 increases forebrain in PANK2^{-/-} mice, which do not possess the neurodegeneration phenotype. I believe data on forebrain CoA levels in PANK1^{-/-} mice is critical to support the concept that the beneficial effects of PZ-2891 are due to increasing brain CoA levels.*

- The CoA measurements in the new knockout mouse model are provided in new Fig. 7c,d. The affected mice had about a 40% reduction in total brain CoA, and PZ-2891 elevated this depressed brain CoA level.
- The experiment on the *Pank2* knockout mice was performed for a different reason. This is the isoform that is missing in PKAN patients, and we wanted to confirm that brain CoA could be elevated when PanK2 activity was absent.

4) *The CoA levels in the brain and liver seem to vary greatly between mouse models. For instance, in Figure 7 the liver CoA levels seem much higher than the liver CoA levels in Figure 6, while the forebrain CoA levels in Figure 7 seem lower than the forebrain CoA in Figure 6. The question arises as to what constitutes a CoA deficiency in these mouse models?*

- Yes, there are age- and strain-dependent differences in the levels of tissue CoA in mice. Age largely accounts for the differences between the measurements in postnatal mice prior to weaning in the experiments with the mouse model in the first submission, and the adult C57Bl6/J mice used in other experiments. However, in the new mouse model the mice reach "adulthood" (6 weeks), and the brain CoA levels (Fig. 7c,d) more closely match the levels seen in adult C57Bl6/J mice (6-8 weeks old) (Fig. 6).
- What constitutes a CoA deficiency in the new mouse model is defined by the data in Fig. 7c, d. In the neuronal *Pank1/Pank2* knockout mice, the development of the phenotype correlates with an approximately 35-40% reduction in brain CoA. In our published

Pank1/Pank2 global knockout study (Garcia et al., 2012), a reduction of >40% in CoA caused a very severe phenotype.

5) *Figure S12: It is proposed that PZ-2871 can restore the CoA deficiency in PANK2^{-/-} mice. However, comparison of CoA levels to WT mice (Figure 6) does not really suggest that a CoA deficiency exists in the PANK2^{-/-} mice.*

- We do not state that there is a CoA deficiency in the *Pank2* knockout mice. There is no brain CoA deficiency and there is no brain-related phenotype. The reason that the experiment was performed was to verify that PZ-2891 would stimulate CoA synthesis in the absence of PanK2 expression. In PKAN patients, PanK2 is absent or defective, and thus its activity would not be present.

6) *Figure 2-5: It is not clear what the sample sizes are in these figures.*

- This information is provided in the figures or their legends. We have reviewed the figure legends to confirm that this information is present, and that we have adhered to the Journal guidelines.

7) *Figure 5: It is not clear what effect pantothenate supplementation has on brain CoA levels in these mice.*

- There is no effect of pantothenate supplementation on tissue CoA levels. In the original figure 5B we had combined the CoA levels in mice that were pantothenate supplemented or not because the values were the same with and without pantothenate. In the revised Fig. 5B, we have broken the data out so that it is obvious that pantothenate itself does not have an impact on CoA levels.

8) *Figure 6i: The increase in hindbrain CoA was less dramatic with higher doses of PZ-2891. Was this due to inhibition of PANK at higher concentrations? Why did this not occur in other tissues such as the liver?*

- Because PZ-2891 is an activator at low concentrations and an inhibitor at high concentrations, one may conjecture that the slightly lower mean CoA levels at the 225 ppm dose compared to the 75 ppm dose in Fig. 6i could indicate that PZ-2891 was approaching an inhibitory concentration at the 225 ppm dose. However, Student's *t* test shows that the difference between the 75 and 225 ppm groups is not significant, so we cannot conclude that there is a difference between these two treatments.

Reviewer #2 (Remarks to the Author):

Here, this referee reports only on the crystallographic part of the manuscript by Lalit Kumar Sharma et al.

The authors describe the crystal structure of a complex of one of the pantothenate kinases (PANK3) from human with AMPPNP and a PANK activator PZ-2891, (PANK3•AMPPNP•Mg²⁺•PZ-2891). In their crystal structure, PZ-2891 binds across the PANK3 dimer interface, in which PZ-2891 occupies the acetyl CoA and the pantothenate binding pockets simultaneously to make a stable PANK3•AMPPNP•Mg²⁺•PZ-2891 complex.

According to the Full wwPDB X-ray Structure Validation Report on the structure (PDB ID : 5TL2), the quality of the structure is sufficiently high. Data and refinement statistics are good. The Fo-Fc simulated annealing omit map (green mesh in fig. S3) looks good. The interactions of PZ-2891 with PANK3 are well presented in Fig. 2.

They soaked the crystals of the PANK3•AMPPNP•Mg²⁺•pantothenate complex with 1 mM PZ-2891 for two days. I wonder if cocrystallization rather than soaking makes any difference in the structure. Do the authors have any opinion on this point?

- We co-crystallized the PANK3•AMPPNP•Mg²⁺•PZ-2891 complex and determined the structure at 2.5 Å resolution with final R_{work} and R_{free} values of 18.9% and 22.4%, respectively. The structures of the complex obtained by co-crystallization and soaking are almost identical. In the figure below, yellow is soaking and magenta is co-crystallization. We continued with the soaking protocol for two reasons; the resolution of the structure is far superior (1.6 Å), and it is much easier to use the soaking method for rapidly determining the structures of many PANK3-pantazine complexes during compound screening and development. The comparison figure below has been incorporated as Supplementary Fig. 4b

Reviewer #3 (Remarks to the Author):

The paper by Sharma et al identified a molecule denominated PZ-2891, which acts as a potent allosteric PANK activator that has oral bioavailability and crosses the blood brain barrier. The study demonstrated that PZ-2891 oral administration to mice increased CoA levels in both the liver and brain, and extended the lifespan in a mouse model of brain CoA deficiency. The Authors claimed this could be a possible therapeutic approach for neurodegenerative diseases caused by mutations of PANK2 gene responsible for PKAN.

The paper is interestingly but there are several weaknesses.

-In the abstract the Authors stated: “PZ-2891, a potent allosteric PANK activator that has oral bioavailability and crosses the blood brain barrier”. Which are the experimental evidences supporting these to relevant aspects? These two points must be further discussed in the manuscript by also citing available literature or original data produced by the Authors.

- PZ-2891 is orally bioavailable based on the pharmacokinetic data presented in Table S2. Also, PZ-2891 was orally administered in all animal experiments, so it is evident that the drug is orally bioavailable. The effect of PZ-2891 on the brain CoA shows that it penetrates the blood brain barrier. To provide additional support for this, we have added new bioanalytical information (new Supplementary Fig. 12) that shows the tissue levels of PZ-2891 in the brain, liver and plasma to directly support this statement. A sentence reporting these results is in the Results section.

-In the Introduction the sentence “The PANK2 gene is abundant in human neuronal tissues and the majority of the mutations associated with PKAN result in the expression of truncated or mutant

PANK2 proteins with little or no catalytic activity” should be modified since there are several mutations associated with increased PANK2 activity. Moreover, Pank2 was also reported in mouse mitochondria by other authors. These papers must be reported and cited in the bibliography.

- The statement is correct as written. In our previous publication, we reported that only 8 out of 22 missense mutants had enzyme specific activities similar to or above wild-type PANK2, i.e. 78-176 % (Zhang et al., J Biol Chem 281:107-114, 2006). Of these ‘active’ mutant proteins, 6 out of 8 mutants are predicted to have reduced protein stability (Hong et al., J Biol Chem 282:27984-27993, 2007) that, in turn, would result in reduced cellular activity.
- We have cited the papers in the revised Introduction where mouse Pank2 was localized to the mitochondria, in addition to the work that shows this is not the case.

-In the results, chapter “PZ-2891 increases intracellular CoA levels in cultured cells”, there is a certain grade of confusion about transfected and un-transfected cells, C3A and 293T cells and it is difficult to follow the data. Why different cell types were used?

- The C3A cells are a human liver lineage cell line that was used to study activation of CoA synthesis in wild-type cells that expressed PANK1, PANK2 and PANK3 at equivalent levels (see Supplementary Figure 9). The HEK293T cells were used in the transfection experiments because these are a standard cell line for overexpression mediated by the pcDNA expression vector. We also think that it is useful to show that the PZ-2891 effect is not specific to one cell line.

Which is the rationale of expressing a mature form of PANK2 in the experiment reported in Fig 4E? This protein can’t reach its “physiological” localization without the mitochondrial targeting signal (MTS). So this experiment should be performed with the PANK2 cDNA containing the MTS to see if the results are different.

- The experiment was designed to evaluate the biochemical activity of the mature PANK2 protein in response to PZ-2891. We were not trying to understand if PANK2 has physiological functions in addition to enzyme activity in mitochondria. The protein was expressed as a cytosolic form to match the other proteins used in the comparative assays. The mature form is the same as the mitochondrial protein except it has been removed from the mitochondria so we can assay it under the same conditions as the other enzymes. PZ-2891 interacts with the processed, mature form of PANK2. See also, new Supplementary Fig. 1 showing data with the purified mature form of human PANK2. The PANK2 protein we are measuring the activity of in both experiments is the form that is present in the mitochondria.

-The last few sentences of this chapter do not take into account several experimental evidences already published:

“CoA itself was reported to raise intracellular CoA levels”: Srinivasan et al (Nat Chem Biol, 2015) demonstrated that CoA is converted into 4PP and is able to rescue several phenotypes in cells and Drosophila genetic models of PKAN. Specifically, CoA is able to rescue a null fumble mutant as well as a hypomorphic mutant, thus indicating it is converted into 4PP and then back to CoA since in Drosophila there is only one PANK and it is not possible for other PANKs to compensate.

- Many of the following Reviewers’ comments were related to Supplemental Fig. 11, and 4-5 lines of related text in the main body of the original version of this manuscript. Here, we provide a Combined Response. We have elected to remove Supplementary Fig. 11 from the paper and the associated text.
- There are several papers that show CoA is neither stable to digestion following oral administration nor stable in extracellular spaces surrounding animal cells or tissues. CoA

and 4'-phosphopantetheine are not incorporated into cells such as *E. coli*, rat liver or neuronal cells (1-3). CoA is degraded to cysteamine and pantothenate in the extracellular space (4). CoA is also degraded to pantothenate by the process of digestion (5). Oral administration of CoA is the same as administering pantothenate plus cysteamine plus adenosine. However, CoA digestion or degradation is not the topic of our paper, and is not related to developing pantazine therapeutics. Rather than provide additional data and text, we have addressed all of the reviewer's concerns by removing the old Supplementary Fig. 11 from the paper along with the text related to it. The removal of this data does not in any way impact the main point of our study, and enables a more concise presentation focused on Pantazines.

- We appreciate that fly and fish model systems can be useful, but we have not discussed these data in the manuscript. There is extensive literature on pantothenate kinase and CoA metabolism in mammals, but CoA metabolism has not been extensively studied in flies and fish, nor have the pantothenate kinases from these species been biochemically characterized. Human therapeutics must be vetted in a mammalian model, and in this regard, the data with fly larvae and fish are not particularly relevant. Our paper does not address the content of these studies, and we have space limitations. Perhaps CoA has an effect in these lower organisms, but what is important for the development of human therapeutics is how is CoA synthesized and metabolized in mammals.

“Pantetheine was not stable in serum (Fig. S11B)”: this was already demonstrated by Srinivasan et al and this ref should be added.

- See Combined Response.

“Although CoA was stable in serum, in the presence of cells it rapidly broke down to pantothenate (Fig. S11C). Orally-administered CoA was not detected in the plasma, but it significantly elevated plasma pantothenate levels (Fig. 11D)”: these data must be supported by biochemical evidence that CoA is directly converted into pantothenate. The Authors should at least measure 4PP to understand if this is an intermediate in this conversion. According to what is known about the biosynthetic pathway of CoA this observation should imply that the pathway can also goes in the backward reaction. A biochemical demonstration supporting this conversion must be provided, since this could be a major and novel achievement.

- See Combined Response.

-“Thus, the conversion of CoA to pantothenate in cells and animals accounts for the reversal of hopantenate toxicity”: as already mentioned there are several additional models demonstrating that CoA can reverse pathological phenotype in genetic models and not only in model of hopantenate toxicity.

- See Combined Response.

Specifically:

*1) Srinivasan et al demonstrated that a *Drosophila* genetic model is rescued by CoA addition and this could be obtained only through CoA conversion into 4PP and not into pantothenate since the only PANK enzyme is absent.*

- See Combined Response.

2) Orellana et al (EMBO Mol Med 2016) demonstrated that CoA was able to rescue pathological phenotypes obtained in neurons directly derived from PKAN patients, a faithful model of disease directly generated from patients' fibroblasts in which expression of PANK2 was abolished by the presence of specific mutations. Furthermore, both PANK2 re-expression in PKAN neurons and

CoA addition to the culture media were able to rescue pathological phenotypes. It is true that pantothenate was not used in these experiments to evaluate its possible rescue capacity, but it is also true that pantothenate is always present in the culture media of neurons. So, a possible scenario would be either that pantothenate should be added in "huge amount" or, if present at adequate concentration, it would have prevented the expression of pathological phenotypes, which are specifically connected to the absence of PANK2.

- These authors showed that CoA partially corrected some of the physiological phenotypes, but they did not show that extracellular CoA increased cellular CoA. The effects may be due to CoA breakdown products like pantothenate, cysteamine (a potent antioxidant), adenine (an autophagy inducer) or adenosine (a signaling molecule), and none of these control conditions were included in the Orenella paper. These controls are needed because CoA is degraded to all of these biologically active molecules. Cysteamine is an excellent antioxidant and ROS scavenger (6-8), and likely accounts for the CoA effect on ROS production. We do not think that we need to discuss this paper, because their results do not measure CoA levels or relate to drug discovery.
- See also combined response.

3) There are also two models of Zebrafish, one for Pank2 (Zizioli et al, 2016) the other for Coasy (Khatri et al, 2016), suggesting that CoA addition is able to rescue pathological phenotypes.

- See Combined Response.

-In figure 5 (B) liver CoA levels in mice maintained on normal diet and treated with 30 mg/kg of PZ-2891 without or with a 200 mg/kg pantothenate supplement were presented.

The Authors should also measure levels of CoA in the brains of the same animals. Moreover, the legend of this figure contains several mistakes.

- The purpose of Fig. 5b was to first determine whether PZ-2891 had an effect in the organ most accessible to the drug (liver) following oral administration, and to determine the pantothenate dependency was observed in cultured cells (Fig. 5a) is also observed in animals. We did not measure brain CoA in this experiment. We do not think that brain CoA data are necessary for this experiment because PZ-2891 is shown to elevate brain CoA levels in multiple experiments that are presented later in the manuscript. These experiments include the dose response to PZ-2891 in six data panels of Fig. 6 in males and females. Also, brain CoA levels increased in response to PZ-2891 in the mouse model in Fig. 7, in Supplementary Fig. 14, and in Pank2 knockout mice In Supplementary Fig. 13. We corrected the figure legend.

It should also be mentioned that plasma pantothenate level was elevated in one PKAN patient taking oral calcium pantothenate as reported by Leoni et al 2012. How could this be reconciled with the observation obtained in wild-type animals in Fig. 5C?

- We do not understand this comment. People who take a pantothenate supplement are expected to have higher plasma pantothenate, which is why the PKAN patient taking pantothenate would have higher pantothenate levels in plasma. In Fig. 5C, the mice fed a pantothenate supplement have higher levels of plasma and tissue pantothenate. There does not seem to be anything to reconcile.

-In Fig S9 again no measurement of CoA levels in the brains is reported. Moreover, it is not clear why phosphopantetheine level is reduced in mice livers but not in the cells. Phosphopantetheine is misspelled in panel C.

- The purpose of these experiments was to determine if PZ-2891 elevates any of the pathway intermediates, other than CoA. The phosphopantetheine is slightly reduced in both cells and liver (compare Supplementary Figure 10c with 10f). We do not know the reason for this small effect. Note the Y-axis labels. Phosphopantetheine is < 1-2% of the CoA. The quantitative distribution of intermediates in the CoA biosynthetic pathway confirms that PanK is the rate-controlling step in CoA biosynthesis. The brain CoA levels are shown in response to doses of PZ-2891 in six panels of Fig. 6, plus Fig. 7 and Supplementary Figures 13 and 14. We fixed the typo in the figure.

-In figure S11 it is compulsory to also measure 4-phosphopantetheine and/or to biochemically demonstrate that CoA is directly converted into pantothenate. I would suggest using a labelled CoA and evaluating by Mass Spec the different intermediates in both the cells and mice.

- See combined response. The results of the suggested experiment were published 5 years ago (4).

-The most interesting part of this manuscript concerned the usage of the double Pank1-/- Syn-Pank2-/- mouse model as a model of CoA deficiency. However, these experiments lack CoA measurement in the liver and brain of PZ-2891 treated animals. The authors should provide these measurements. Moreover, they should explain why animals die after 26 days of treatment, even though two animals survive longer. Again, in this last case it is not clear if after 66 or 119 days of treatment mice were euthanized or spontaneously died. It is important to discuss this point in light of what was previously reported by Corbin et al 2017, that excess of CoA level reduces skeletal muscle performance and strength. Do these mice die because of respiratory distress due to diaphragm insufficiency?

- We have provided an entire series of experiments with the next-generation mouse model as the new Fig. 7. Importantly, these animals live longer and have a severe locomotion defect. The treated animals have improved locomotor performance. Please also see our response to Reviewer 1.
- We measured the impact of PZ-2891 on heart and muscle tissue CoA in animals treated for one week on the drug and included these data as new Supplementary Fig. 14. PZ-2891 did not increase CoA in skeletal muscle. The experiments of Corbin and the current experiments are different. Corbin significantly increased PanK levels in skeletal muscle as mediated by *Pank2* overexpression driven by a transgene, which in turn elevated CoA. They also report that muscle has a lower level of PanK isoform mRNA expression than other tissues. In the current studies, PZ-2891 can only activate the amount of PanK enzyme that is present in each tissue, and we observe little impact on muscle CoA levels. The fact that PZ-2891 therapy improves the locomotor activity of neuron-selective *Pank1/Pank2* double knockout mice, and the lack of any detrimental effect of PZ-2891 on the locomotor activity of wild-type mice, provides additional strong evidence that pantazines do not significantly affect muscle performance (Fig. 7 & Supplementary Figs. 16, 17). Note that brain CoA was elevated in these wild-type cohorts of mice.

References

1. Balibar, C. J., Hollis-Symynkywicz, M. F., and Tao, J. (2011) Pantethine rescues phosphopantothenoylcysteine synthetase and phosphopantothenoylcysteine decarboxylase deficiency in *Escherichia coli* but not in *Pseudomonas aeruginosa*. *J. Bacteriol.* **193**, 3304-3312
2. Domschke, W., Liersch, M., and Decker, K. (1971) Lack of permeation of coenzyme A from blood into liver cells. *Hoppe Seylers Z Physiol. Chem.* **352**, 85-88

3. Kropf, M., Rey, G., Glauser, L., Kulangara, K., Johnsson, K., and Hirling, H. (2008) Subunit-specific surface mobility of differentially labeled AMPA receptor subunits. *Eur. J. Cell Biol.* **87**, 763-778
4. Wu, J., Sandberg, M., and Weber, S. G. (2013) Integrated electroosmotic perfusion of tissue with online microfluidic analysis to track the metabolism of cystamine, pantethine, and coenzyme A. *Anal. Chem.* **85**, 12020-12027
5. Shibata, K., Gross, C. J., and Henderson, L. M. (1983) Hydrolysis and absorption of pantothenate and its coenzymes in the rat small intestine. *J. Nutr.* **113**, 2107-2115
6. Aruoma, O. I., Halliwell, B., Hoey, B. M., and Butler, J. (1988) The antioxidant action of taurine, hypotaurine and their metabolic precursors. *Biochem. J.* **256**, 251-255
7. Ferreira, D. W., Naquet, P., and Manautou, J. E. (2015) Influence of vanin-1 and catalytic products in liver during normal and oxidative stress conditions. *Curr. Med. Chem.* **22**, 2407-2416
8. Lv, W., Booz, G. W., Fan, F., Wang, Y., and Roman, R. J. (2018) Oxidative stress and renal fibrosis: Recent insights for the development of novel therapeutic strategies. *Front. Physiol.* **9**, 105
9. Zhang Y-M, Chohnan S, Virga KG, Stevens RD, Ilkayeva OR, Wenner BR, Bain JR, Newgard CB, Lee RE, Rock CO, Jackowski S. (2007) Chemical knockout of pantothenate kinase reveals the metabolic and genetic program responsible for hepatic coenzyme A homeostasis. *Chem & Biol* 14:291-302.

Response to Review

Reviewers' comments:

Reviewer #1 (Remarks to the Author):

None

Reviewer #2 (Remarks to the Author):

The authors have addressed my comment perfectly, and I have no objection against the acceptance of this manuscript.

Reviewer #3 (Remarks to the Author):

The Authors have addressed the majority of the questions raised in the first revision. The generation of the novel animal model is a great achievement and represents an excellent platform to test therapeutic approaches for PKAN.

However, I kindly disagree with some of their answers:

- The experiment was designed to evaluate the biochemical activity of the mature PANK2 protein in response to PZ-2891. We were not trying to understand if PANK2 has physiological functions in addition to enzyme activity in mitochondria. The protein was expressed as a cytosolic form to match the other proteins used in the comparative assays. The mature form is the same as the mitochondrial protein except it has been removed from the mitochondria so we can assay it under the same conditions as the other enzymes. PZ-2891 interacts with the processed, mature form of PANK2. See also, new Supplementary Fig. 1 showing data with the purified mature form of human PANK2. The PANK2 protein we are measuring the activity of in both experiments is the form that is present in the mitochondria.*

This experiment implies that the overexpressed PANK2 protein works in the cytosol and not in the mitochondria, where it is physiologically located. So the measurement in figure 4e could be not correct because of the wrong localization of the PANK2 protein. I would suggest removing the data on PANK2 or perform the correct experiment. Moreover, the novel revised figures 4d and 4e do not contain the legend of the color's bars.

First, we have accepted the suggestion of the reviewer to simply remove the PANK2 bar from Fig. 4e. Note that this means that we have cut the lanes in the gel below the figure to remove the PANK2 lane from the western blot. Second, we have made sure it is clear what the colors indicate in the revised Fig. 4d and 4e. The PANK2 data in Fig. 4E was not specifically pointed out in the text, so it is not needed. PANK2 has been struck from the text and Figure 4e legend, and revised Figs. 4d and 4e are provided.

In the Introduction they just mentioned:

“phosphopantetheine, CoA, or S-acetyl-phosphopantetheine have been suggested as PKAN modulators based on their ability to reverse hopantenate inhibition of coA synthesis...”

This information is not complete since papers indicating that CoA is able to rescue other genetics models (human neurons and Zebrafish) of disease are available. The request was just to cite the available literature and not to discuss the scientific content.

Introduction: we have modified the text by inserting the exact wording suggested by the reviewer and have added an additional 5 references on the correction of CoA deficiencies in neurons and lower organisms. These are references 24-30.